# Herbivory-induced volatiles function as defenses increasing fitness of the native plant *Nicotiana attenuata* in nature

**Meredith C Schuman, Kathleen Barthel[†], Ian T Baldwin***

Department of Molecular Ecology, Max Planck Institute for Chemical Ecology, Jena, Germany

**Abstract** From an herbivore's first bite, plants release herbivory-induced plant volatiles (HIPVs) which can attract enemies of herbivores. However, other animals and competing plants can intercept HIPVs for their own use, and it remains unclear whether HIPVs serve as an indirect defense by increasing fitness for the emitting plant. In a 2-year field study, HIPV-emitting *N. attenuata* plants produced twice as many buds and flowers as HIPV-silenced plants, but only when native *Geocoris* spp. predators reduced herbivore loads (by 50%) on HIPV-emitters. In concert with HIPVs, plants also employ antidigestive trypsin protease inhibitors (TPIs), but TPI-producing plants were not fitter than TPI-silenced plants. TPIs weakened a specialist herbivore's behavioral evasive responses to simulated *Geocoris* spp. attack, indicating that TPIs function against specialists by enhancing indirect defense.

***For correspondence:** baldwin@ice.mpg.de

**†Present address:** Federal Research Center for Cultivated Plants, Institute for Breeding Research on Horticultural And Fruit Crops, Julius Kühn Institute, Dresden, Germany

**Reviewing editor**: Detlef Weigel, Max Planck Institute for Developmental Biology, Germany

## Introduction

Plant indirect defenses are traits that disable or remove herbivores by manipulating tri-trophic interactions to the advantage of the plant (*Price et al., 1980*). They attract and inform the third trophic level, predators or parasitoids, resulting in increased attacks on herbivores (*Turlings and Wäckers, 2004*). Indirect defenses are widespread and include domatia, extrafloral nectar, and food bodies which provide shelter and nutrition for predators and parasitoids, as well as herbivory-induced plant volatiles (HIPVs) which convey information about feeding herbivores (*Heil, 2008*). Field studies with the native tobacco *Nicotiana attenuata*, a desert annual, and with maize have shown that HIPVs can reduce herbivore loads by 24% to more than 90%, by increasing both predation and parasitization of herbivores (*Kessler and Baldwin, 2001*; *Rasmann et al., 2005*; *Halitschke et al., 2008*; *Degenhardt et al., 2009*; *Allmann and Baldwin, 2010*) and deterring herbivore oviposition (*Kessler and Baldwin, 2001*).

If HIPVs really function as defenses, they should increase Darwinian fitness, defined as successful reproduction, for plants under herbivore attack (*Karban and Baldwin, 1997*). But because HIPVs can be perceived by many other members of the ecological community—from herbivores, pollinators, predators and parasitoids to competing or parasitic plants—it is not clear whether HIPVs increase plant fitness in nature (*Dicke and Baldwin, 2010*; *Kessler and Heil, 2011*). The field studies described above have either spanned too short a time to reveal Darwinian fitness benefits, or have not reported fitness data at all (*Kessler and Baldwin, 2001*; *Rasmann et al., 2005*; *Halitschke et al., 2008*; *Degenhardt et al., 2009*; *Allmann and Baldwin, 2010*). Two laboratory studies showed that parasitization of herbivores can increase plant reproduction (*van Loon et al., 2000*; *Hoballah and Turlings, 2001*), but the parasitization in these studies was not mediated by HIPVs. Hence three decades after their description, it remains unclear whether HIPVs are really indirect defenses.

Long-term field studies comparing HIPV-emitting vs -deficient plants are required in order to demonstrate a defensive function for HIPVs. Experimental additions of pure volatiles or mixes to

**eLife digest** As the population of the world continues to increase beyond 7 billion, and agricultural pests continue to rapidly evolve resistance to pesticides, it is becoming ever more important to cultivate arable land in a way that is sustainable for both humans and the environment. A better understanding of the different mechanisms used by wild plants to deter herbivores will help to increase crop production without harming the environment.

Plants use both direct and indirect methods to fend off herbivores. Direct defense methods include the production of chemicals that are toxic to herbivores or give them indigestion, and the growth of sticky prickles and spines that can injure or kill the herbivore. Indirect defense methods, on the other hand, generally rely on the plant attracting organisms that are either predators or parasites of the herbivore.

Plants produce odors known as herbivory-induced plant volatiles (HIPVs) that are thought to offer indirect defense against herbivores by betraying their location to predators and parasites. However, HIPVs also influence other members of the ecological community, sometimes in ways that are detrimental to plants. Moreover, despite 30 years of research, no study has demonstrated that HIPVs increase the fitness of a plant, so it is unclear what they have evolved to do.

Now, a 2-year field study by Schuman et al. has shown plants that emit green leaf volatiles (which are a type of HIPV) produce twice as many buds and flowers—a measure of fitness—as plants that have been genetically engineered not to emit green leaf volatiles. This study was conducted with *Nicotiana attenuata*, which is a wild tobacco plant that is often targeted by *Manduca sexta*, a type of moth that is also known as the tobacco hornworm. Green leaf volatiles only increased plants' fitness when various species of *Geocoris*—a bug that preys on *Manduca sexta*—reduced the number of herbivores by a factor of two. This is the first evidence that HIPVs offer indirect defense against herbivores.

Schuman et al. also studied the effects of molecules called protease inhibitors that are thought to function as direct defenses by making it difficult for herbivores to digest plants. They found that the ability to produce protease inhibitors did not increase the fitness of plants under herbivore attack; however, tobacco hornworms that had been fed plants containing protease inhibitors were found to be more sluggish in response to attack, which suggests that protease inhibitors can enhance the indirect defenses of plants. The results suggest that employing both direct and indirect defenses—such as a combination of biological pesticides and genetic engineering to produce both HIPVs and protease inhibitors—is the best approach for defending agricultural plants against pests.

plants growing in nature has worked well to test short-term effects of specific compounds (***Kessler and Baldwin, 2001***; ***Allmann and Baldwin, 2010***), but only endogenously produced HIPV emissions can ensure specific, lasting and consistent differences under field conditions. Most evidence for the utility of HIPVs comes from studies in which predators and parasitoids learn to associate HIPVs with prey; naïve predators and parasitoids are just as likely to respond to HIPVs as not to respond (***Allison and Hare, 2009***). Thus the inducibility of HIPV emission, which ensures association with herbivore feeding, is likely essential for HIPV function, but it is difficult to engineer (***Kos et al., 2009***). Engineered constitutive HIPV emissions have been used, either on predators and parasitoids trained to associate target volatiles with prey in short-term laboratory experiments (***Kappers et al., 2005***; ***Schnee et al., 2006***), or in set-ups in which target volatiles are always associated with prey (***Rasmann et al., 2005***; ***Degenhardt et al., 2009***). When plants are engineered constitutively to emit HIPVs, they no longer provide accurate information about the location of feeding herbivores, and predators will not associate these signals with prey in nature. Genetically silencing the biosynthesis of HIPVs, however, permits naturally inducible wild-type (WT) plants to serve as HIPV emitters, for comparison with transformed lines lacking specific volatile components (***Halitschke et al., 2008***; ***Skibbe et al., 2008***). Furthermore, field experiments that manipulate the production of HIPVs which not only attract the third trophic level, but also influence the second trophic level (e.g., as feeding stimulants and host location cues), require additional experimental manipulations to preserve the plant-herbivore part of the tritrophic interaction.

When HIPVs do attract the third trophic level, how can herbivores adapt? Many herbivores can outgrow their vulnerability to predators and parasitoids, but plant direct defenses can slow herbivore growth and prolong vulnerability as postulated by the slow growth-high mortality hypothesis (*Benrey and Denno, 1997*; *Williams, 1999*; *Kessler and Baldwin, 2001*, *2004*; *Kaplan and Thaler, 2011*). The solanaceous specialists *Manduca sexta* and *M. quinquemaculata* (Lepidoptera, Sphingidae) are resistant to the potent alkaloid toxin nicotine (*Wink and Theile, 2002*), but sensitive to the nutritional value of plant tissue (*Zavala and Baldwin, 2004*). Non-toxic protease inhibitor (PI) proteins, which inhibit protein digestion and thus decrease the availability of organic nitrogen in the form of amino acids (*Zavala et al., 2008*), are widespread in flowering plants (*Hartl et al., 2011*), and trypsin protease inhibitors (TPIs) slow the growth of *M. sexta* on *N. attenuata* (*Zavala et al., 2008*). However, herbivores can overcome PIs by producing insensitive or desensitized proteases, inactivating or degrading PIs, eating more plant tissue, and eating more nutritious young tissue (*Winterer and Bergelson, 2001*; *Steppuhn and Baldwin, 2007*; *Zavala et al., 2008*). In the latter two cases, PIs could reduce plant fitness. Although TPI-producing *N. attenuata* plants produce more seeds than TPI-deficient plants when attacked by *M. sexta* under controlled glasshouse conditions (*Zavala and Baldwin, 2004*), whether TPIs function as a direct defense in nature is unknown.

We tested the hypotheses that HIPVs and TPIs defend plants in nature by increasing herbivore predation and thereby plant Darwinian fitness. To do so, we monitored the performance, predation and mortality of *Manduca* spp. (*M. sexta* and *M. quinquemaculata*) on wild-type *N. attenuata* plants and RNAi transformed lines silenced for the production either of a specific group of HIPVs, or of TPIs, and compared the resulting plant reproductive output in terms of bud and flower production (we are not permitted to allow transgenic plants to disperse ripe seed). Because *N. attenuata* is an annual, opportunistic out-crosser, seeds are produced within one growing season, mostly from fertilization via self-pollen (*Sime and Baldwin, 2003*), and we can relate bud and flower production to lifetime seed production, which is commonly accepted as a measure of Darwinian fitness (*Baldwin, 1998*; *van Loon et al., 2000*; *Hoballah and Turlings, 2001*).We hypothesized that HIPVs would increase plant reproduction by increasing predation of herbivores, and that TPIs alone would not increase reproduction under herbivore attack, but would either increase predation or increase herbivores' susceptibility to predators. We then assembled a toolbox of wild-type and transgenic lines chosen to test these hypotheses.

We chose a genotype of *N. attenuata* native to the Great Basin Desert of southwestern Utah. In many years, *Manduca* spp. larvae cause the most defoliation of *N. attenuata* plants in this area (*Kessler and Baldwin, 2001*) and thus the *N. attenuata* 'UT' genotype is likely adapted to defend against *Manduca* spp. Eggs and young larvae of *Manduca* spp. are predated by *Geocoris* spp. (big-eyed bugs) which occur naturally in the Utah habitat and are attracted to components of *N. attenuata*'s HIPV bouquet (*Kessler and Baldwin, 2001*; *Halitschke et al., 2008*; *Skibbe et al., 2008*). Specifically, Utah *Geocoris* spp. predators are attracted to the sesquiterpene (*E*)-α-bergamotene as well as green leaf volatiles (fatty acid-derived $C_6$ aldehydes, alcohols and esters) (*Kessler and Baldwin, 2001*; *Halitschke et al., 2008*; *Schuman et al., 2009*). Green leaf volatiles, or GLVs, can be silenced via a single upstream 13-lipoxygenase, Na*LOX2*, which specifically supplies lipid hydroperoxides for their production (*Allmann et al., 2010*). Although GLVs are released upon mechanical damage, the oral secretions (OS) of *M. sexta* convert 3-(*Z*)-GLVs to the 2-(*E*)-structures, resulting in greater *Geocoris* spp. predation than the damage-induced (*Z*):(*E*) ratio (*Allmann and Baldwin, 2010*). GLVs are released immediately upon damage (*Allmann and Baldwin, 2010*) and may therefore be a 'first line of defense'.

Like GLVs, many other HIPVs are also released after mechanical damage, but change in amount or ratio upon herbivory, and thus GLVs mirror the functional complexity of the total HIPV blend. Furthermore, GLVs prime or directly regulate responses in neighboring plants (*Kessler et al., 2006*; *Paschold et al., 2006*), attract herbivores as well as predators (*Halitschke et al., 2008*), and are important cues for pollinating and ovipositing moths (*Kessler and Baldwin, 2001*, *2006*; *De Moraes et al., 2001*; *Fraser et al., 2003*), thus performing several roles which may harm or benefit plant fitness in addition to their role in attracting predators. Perhaps most significantly, GLVs also stimulate *Manduca* spp. feeding, and silencing plant GLV production results in reduced herbivore damage (*Halitschke et al., 2004*; *Meldau et al., 2009*). All these qualities made the manipulation of GLV emissions an ideal means to test rigorously the fitness consequences of HIPV emissions and to evaluate whether these emissions can truly be considered defensive.

## Results

### GLV and TPI production are reduced or eliminated in transformed lines

We chose a line of ir*PI* plants with no detectable TPI activity (**Steppuhn and Baldwin, 2007**), and a line of ir*LOX2* plants with GLV emissions <20% of WT (**Allmann et al., 2010**); non-target defense metabolites are not affected in either line (**Steppuhn and Baldwin, 2007**; **Allmann and Baldwin, 2010**), including emission of (*E*)-α-bergamotene measured in a glasshouse characterization of all lines prior to field release (see 'Non-target metabolites are not affected in ir*LOX2*, hemi-ir*LOX2* or ir*PI* plants'). Because of the importance of GLVs for the plant-herbivore interaction, we used both homozygous (**Allmann et al., 2010**) and hemizygous ir*LOX2* plants to provide different levels of GLV silencing. Hemizygous (hemi-) ir*LOX2* plants were created by crossing homozygous ir*LOX2* and ir*PI* plants, but the ir*PI* construct was not active in this cross (**Figure 1** see 'Discussion').

The ir*PI* plants (**Steppuhn and Baldwin, 2007**) had no detectable TPI activity in the glasshouse or throughout the field experiment in 2011, and *PI* transcripts accumulated to only 0.3% of WT levels in ir*PI* (transcripts, N=5, p<0.001 in Scheffe *post hoc* tests following two-way ANOVAs on $log_2$-transformed data with factors W+OS treatment and genotype: treatment $F_{1,29}$=75.909, p<0.001; genotype $F_{3,29}$=174.077, p<0.001); in contrast, TPI activity and *PI* transcripts were similar to WT plants in ir*LOX2* and hemi-ir*LOX2* (transcripts, N=5, p>0.2 in Scheffe *post hoc* tests following two-way ANOVAs on $log_2$-transformed data with factors W+OS treatment and genotype; activity, N=10–17, p>0.05 in one-way ANOVAs with factor genotype) (**Figure 1**).

We assessed GLV production by hexane extraction of GLVs from frozen leaf tissue, and GLV emission by GC analyses of leaf headspaces. GLVs in hemi-ir*LOX2* plants were reduced to levels similar to those in ir*LOX2* plants, but hemi-ir*LOX2* plants still produced detectable amounts of (*Z*)-3-hexenol (**Figures 2 and 3**). The dominant GLV in hexane tissue extracts was (*E*)-hex-2-enal, and (*Z*)-hex-3-en-1-ol was additionally quantifiable as a minor component. Only (*E*)-hex-2-enal was quantifiable in extracts from field-grown plants on May 28, 2011, and was below quantifiable levels in ir*LOX2* and hemi-ir*LOX2* plants, but detectable in pooled samples from hemi-ir*LOX2* (**Figures 2 and 3**). Extracts from later in the season also contained quantifiable amounts of (*Z*)-hex-3-en-1-ol and hemi-ir*LOX2* extracts contained up to 50% as much of this alcohol as WT and ir*PI* extracts (N=10, p<0.05 in Scheffe *post hoc* tests following one-way ANOVAs with factor genotype: June 14, 2011, $F_{2,26}$=9.556, p=0.001; June 22, 2011, $F_{2,26}$=12.196, p<0.001; p>0.6 for ir*PI* vs WT in a t-test for May 28 and in Scheffe *post hoc* tests for June 14 and 22) (**Figure 3**). Headspace measurements from field- and glasshouse-grown plants detected a similar 80–100% reduction in GLV emissions from ir*LOX2* and hemi-ir*LOX2* plants compared to WT and ir*PI* (field, N=3, p=0.024 for hemi-ir*LOX2* v ir*PI*, p>0.05 for hemi-ir*LOX2* vs WT and ir*LOX2* vs ir*PI* and WT, but p=0.939 for ir*PI* vs WT in Scheffe *post hoc* tests following one-way ANOVA: $F_{3,8}$=7.346, p=0.011; glasshouse, N=4: ir*LOX2* and hemi-ir*LOX2* below limit of detection, p=0.834 for t-test ir*PI* vs WT) (**Figure 3**), and transcript accumulation of *LOX2* was 2% of WT levels in ir*LOX2* and hemi-ir*LOX2* (N=5, p<0.001 in Scheffe *post hoc* tests following two-way ANOVAs on $log_2$-transformed data with factors W+OS treatment and genotype: treatment $F_{1,32}$=0.021, p=0.887; genotype $F_{3,32}$=635.477, p<0.001) but unaffected in ir*PI* (p>0.9 in Scheffe *post hoc* test vs WT) (**Figure 3**).

### Non-target metabolites are not affected in transformed lines

For the 'UT' genotype of *N. attenuata* used in our experiments, the induction of all HIPVs *except GLVs* is mediated by jasmonate signaling (**Halitschke and Baldwin, 2003**; **Kessler et al., 2004**). The ir*PI* line A-04-186-1 (**Steppuhn and Baldwin, 2007**) and ir*LOX2* line A-04-52-2 (**Allmann et al., 2010**) have been characterized previously, and neither is affected in jasmonate signaling. Particularly, the emission of (*E*)-α-bergamotene, the best-characterized HIPV in *N. attenuata* apart from GLVs (**Halitschke et al., 2000**; **Kessler et al., 2004**; **Halitschke et al., 2008**; **Skibbe et al., 2008**), does not differ significantly among the lines used (N=4 measured 24–32 hr after W+OS treatment as according to **Halitschke et al. (2000)** and normalized as a percentage of the internal standard peak: WT, 67.9±17.1%; ir*PI*, 30.2±13.2%; ir*LOX2*, 26.6±5.8%; hemi-ir*LOX2*, 42.7±23.3%; ANOVA: $F_{3,12}$=1.338, p=0.308). The transformation process itself does not affect plant fitness or competitive ability (**Schwachtje et al., 2008**), TPI production or volatile emission (**Figures 1–3**).

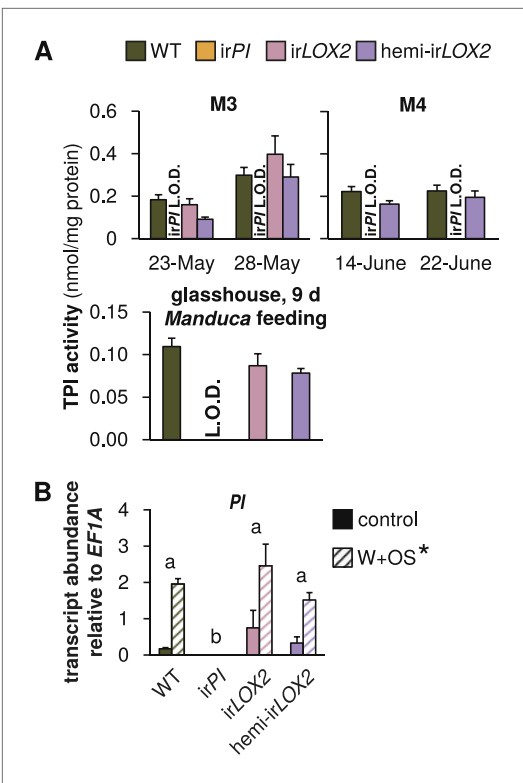

**Figure 1**. Trypsin protease inhibitor (TPI) activity and transcripts in transformed lines; graphs show means+SEM. (**A**) TPI activity measured in systemic leaves of field-grown (top two panels, 2011, N=11–14 for panel 1 and N=21 for panel 2) or glasshouse-grown plants (bottom panel, N=10) attacked by *Manduca* spp. larvae. Only WT, ir*PI* and hemi-ir*LOX2* plants were used in M4. For a timeline of *Manduca* spp. infestations M1–M4 see **Figure 4A**. For raw data, see F2A_SchumanBarthelBaldwin2012TPIactivity.xlsx (Dryad: **Schuman et al., 2012**). (**B**) Transcripts of *PI* in unelicited leaf tissue (control), and at the point of maximum accumulation in W+OS-elicited leaf tissue in glasshouse-grown plants (N=5). For raw data, see F2B_SchumanBarthelBaldwin2012PItranscripts.xlsx (Dryad: **Schuman et al., 2012**). *W+OS treatment had a significant effect on *PI* (p<0.001) transcript accumulation. [a, b] Different letters indicate significant differences between genotypes (p<0.001) in Scheffe *post hoc* tests following a two-way ANOVA on log$_2$-transformed data with factors treatment and genotype (genotype F$_{3,29}$=174.077, p<0.001; treatment F$_{1,29}$=75.909, p<0.001). L.O.D.: below limit of detection for measurement.

## *Geocoris* spp. consistently prefer to predate from GLV-perfumed or -emitting plants

We monitored the predation of *Manduca* spp. larvae and eggs daily, and counted *Geocoris* spp. individuals around plants every 2–3 days (*Geocoris* spp. counts, **Figure 4**). In 2010, we planted into a first-year plot. Although plants were infested with laboratory strain *M. sexta* larvae (N=51) and baited with *M. sexta* eggs (N=50) over a 2-week period in 2010 (**Figure 4**), no *Geocoris* spp. individuals were observed on this plot through May. There were also no *Geocoris* spp. observed through May on a nearby, older plot: *Geocoris* spp. first arrived and began to predate *Manduca* spp. eggs on the older plot on June 9. In 2011, we planted into the older plot, where we observed *Geocoris* spp. in May prior to the first infestation (M2, **Figure 4**).

During infestation M2 (**Figure 4**), we allowed *Geocoris* spp. to associate all four plant genotypes with the presence of prey: we infested half of all plants with equal numbers of first-instar *M. sexta* larvae from the laboratory strain and, because *Geocoris* spp. predate more from GLV-emitting or-perfumed plants (**Kessler and Baldwin, 2001**; **Halitschke et al., 2008**; **Allmann and Baldwin, 2010**), we supplemented GLV emission from ir*LOX2* and hemi-ir*LOX2* plants by placing cotton swabs with lanolin paste containing GLVs representative of the *M. sexta*-fed *N. attenuata* headspace (**Table 1 Allmann and Baldwin, 2010**) adjacent to *M. sexta*-infested leaves. Swabs containing lanolin with solvent were placed next to ir*PI* and WT as a control. *M. sexta* larvae were predated at a rate of 12–37% over two 2- to 3-day trials. *Geocoris* spp. tended to predate more larvae from GLV-supplemented plants (Fisher's exact tests, 35–37% vs 22–27% May 5–6, N=59–60 larvae, p=0.066; 17–21% vs 12% May 13–15, N=92–100 larvae, p=0.069; combined trials, Bonferroni-corrected p=0.0063) (**Figure 5**).

We removed the cotton swabs and the remaining larvae. We then monitored predation of newly-infested larvae and eggs *without* GLV supplementation during infestation M3 (**Figure 4**). We staggered infestation to accommodate differences in plant growth: WT and ir*PI* seedlings were initially larger and therefore were planted into the field on average 3 days earlier than ir*LOX2* and hemi-ir*LOX2* plants, so that all plants were planted at a similar size, which is important for even establishment. We therefore re-infested WT and ir*PI* plants earlier after M2, to allow ir*LOX2* and hemi-ir*LOX2* plants to catch up in their growth to WT and ir*PI* before re-infestation, so as not to bias further assays. However, we left *M. sexta* larvae on ir*LOX2* and hemi-ir*LOX2* as long as on WT and ir*PI*, and we made several control measurements to ensure that differences in *Geocoris* spp. predation were not due to our staggering of infestation: we counted *Geocoris* spp. populations around all genotypes over this period (**Table 2**) and saw that they

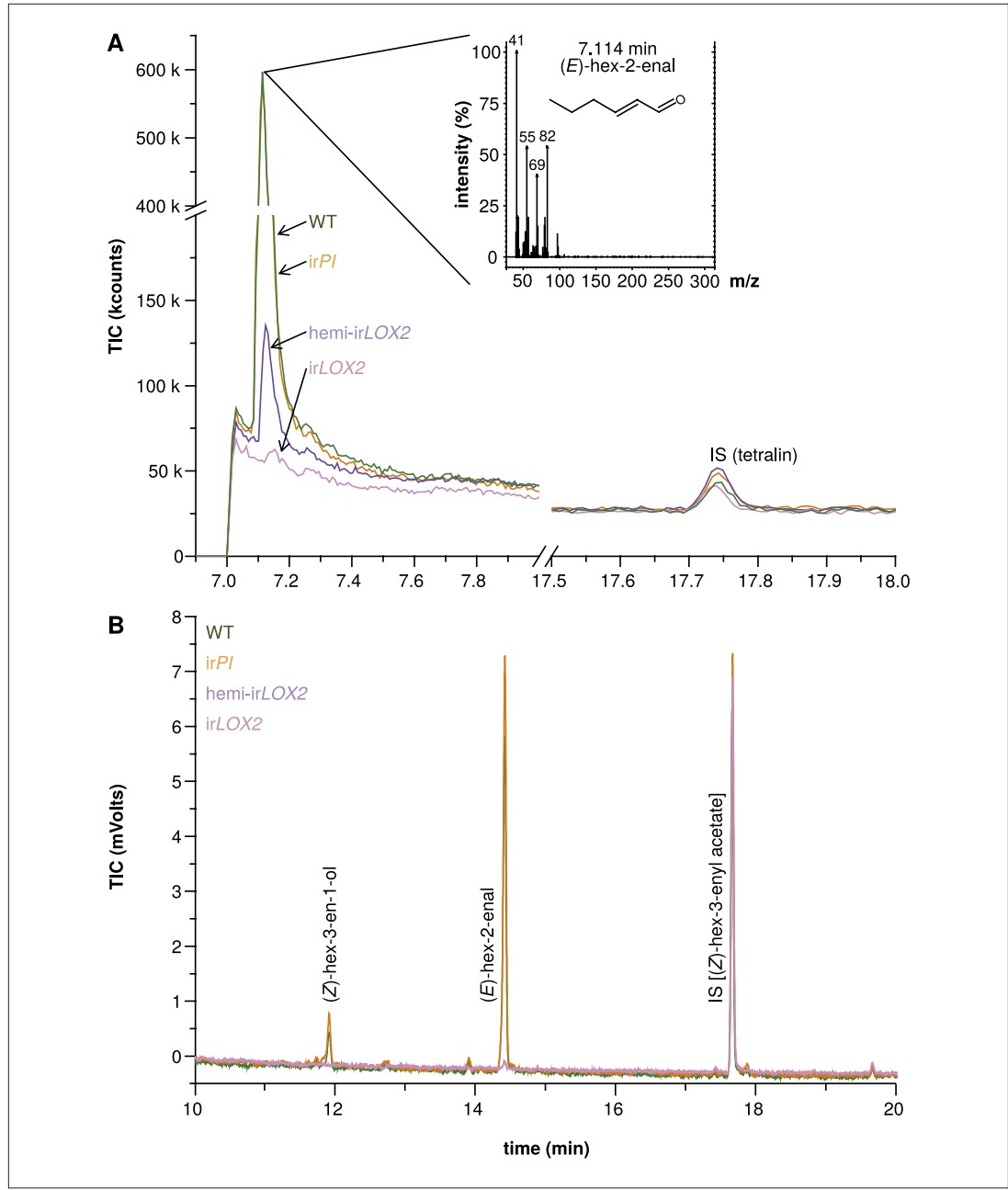

**Figure 2**. Hexane extracts of leaves from field-grown plants. (**A**) Hexane extracts from pooled leaf samples of field-grown plants for a qualitative assessment of green leaf volatile (GLV) pools, analyzed by GC-MS with a split ratio of 1/100 onto a nonpolar column; only (*E*)-hex-2-enal was identified due to poor resolution of (*E*)-hex-2-enal and (*Z*)-hex-3-en-1-ol on the nonpolar column; no ester peaks were detected. For raw data, see F3A_SchumanBarthelBaldwin2012chromatograms.xlsx (Dryad: ***Schuman et al., 2012***). (**B**) Example chromatograms from hexane extracts of individual leaf samples from field-grown plants, analyzed by GC-FID on a wax column. The dominant compound was (*E*)-hex-2-enal; (*Z*)-hex-3-en-1-ol was also present in quantifiable amounts. (*Z*)-3-hexenyl acetate was chosen as an internal standard because no esters were detectable in the preliminary qualitative GC-MS analysis (1A), and because its chemical similarity to (*E*)-hex-2-enal and (*Z*)-hex-3-en-1-ol made it a good choice of internal standard for normalization and calculation of yield from extracts. For raw data, see F3B_SchumanBarthelBaldwin2012chromatograms.xlsx (Dryad: ***Schuman et al., 2012***). IS: internal standard.

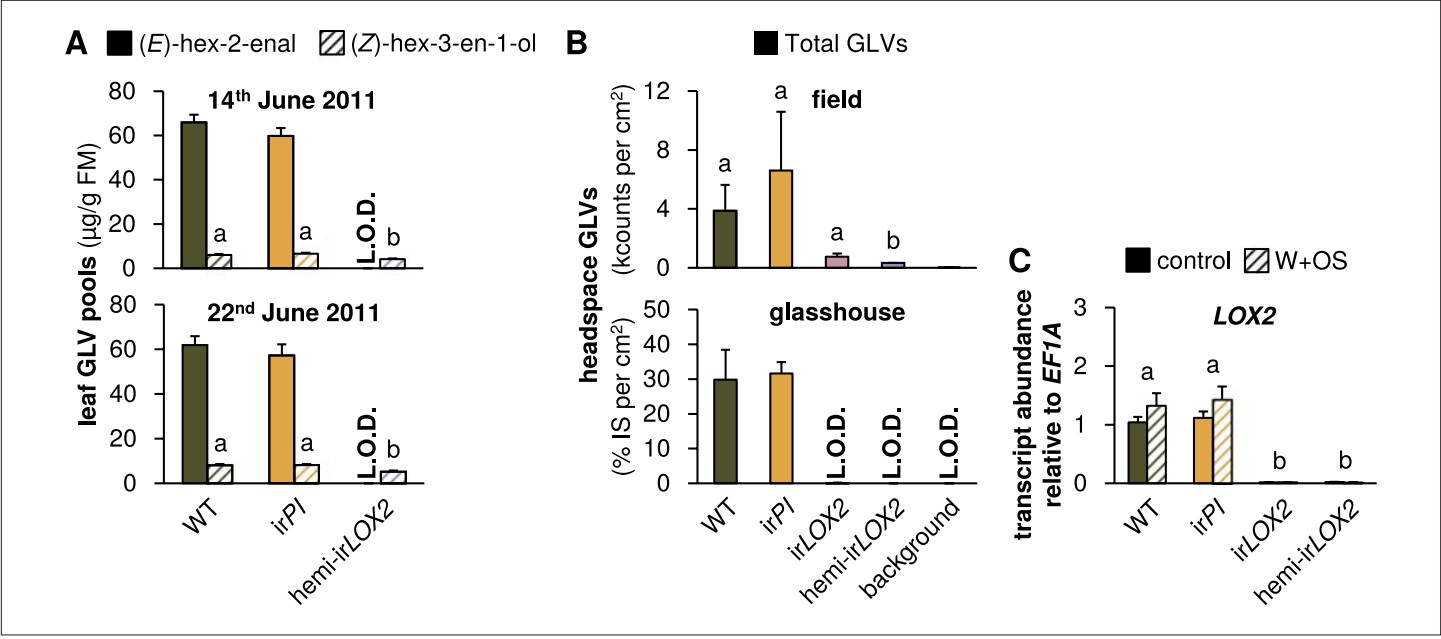

**Figure 3**. GLV production and emission in transformed lines; graphs show means+SEM. (**A**) GLVs extracted with hexane from leaf tissue of field-grown WT, ir*PI*, and hemi-ir*LOX2* plants grouped in triplets for infestation M4 in 2011(***Figure 4A***). Leaves were harvested from every plant at the beginning (June 14) and in the middle of M4 (June 22) and leaves from plants in 10 randomly chosen triplets were analyzed. Only (*E*)-hex-2-enal and (*Z*)-hex-3-en-1-ol were quantifiable in leaf extracts. Different letters (a and b) indicate significant differences (p≤0.05) in Scheffe *post hoc* tests following one-way ANOVAs for (*Z*)-hex-3-en-1-ol (top panel, $F_{2,26}$=9.556, p=0.001; bottom panel, $F_{2,26}$=12.196, p<0.001). For raw data, see F4A_SchumanBarthelBaldwin2012GLVpools.xlsx (Dryad: ***Schuman et al., 2012***). (**B**) GLVs measured in headspace samples of leaves from field-grown (top panel, N=3) or glasshouse-grown plants (bottom panel, N=4). For field-grown plants, leaves were harvested and measured on May 21 (just before M3). Intact leaves were kept fresh by placing petioles in water. Immediately before each measurement, one leaf was treated with wounding and *M. sexta* oral secretions (W+OS); a 1-cm² disc was stamped out and placed in a 4-mL GC vial. After 15 min the headspace in the vial was measured with a Z-Nose 4200 and total alcohols and aldehydes were quantified. Different letters (a and b) indicate significant differences (p<0.05) in Scheffe *post hoc* tests following one-way ANOVA ($F_{3,8}$=7.346, p=0.011). For glasshouse-grown plants, leaves were left on plants, treated with W+OS, and enclosed in padded, 50 mL food-quality plastic containers for 3 hr while the headspace was pulled over a Poropak Q filter. Filter eluents were measured by GC-MS. Three-hour headspace samples contained (*Z*)-hex-3-en-1-ol, (*E*)-hex-2-en-1-ol (forms from (*E*)-hex-2-enal on filters over trapping periods longer than 20 min), (*Z*)-hex-3-enyl acetate, (*Z*)-hex-3-enyl butanoate, (*Z*)-hex-3-enyl isobutyrate, and (*Z*)-hex-3-enyl propanoate, all of which showed the pattern shown for the total amount. For raw data, see F4B_SchumanBarthelBaldwin2012GLVheadspace.xlsx (Dryad: ***Schuman et al., 2012***). (**C**) Transcripts of *LOX2* in unelicited leaf tissue (control), and at the point of maximum accumulation in W+OS-elicited leaf tissue in glasshouse-grown plants (N=5). For raw data, see F4C_SchumanBarthelBaldwin2012LOX2transcripts.xlsx (Dryad: ***Schuman et al., 2012***). [a, b] Different letters indicate significant differences between genotypes (p<0.001) in Scheffe *post hoc* tests following a two-way ANOVA on log₂-transformed data with factors treatment and genotype (genotype $F_{3,32}$=635.477, p<0.001, treatment $F_{1,32}$=0.021, p=0.887). L.O.D.: below limit of detection for measurement.

were not different (p>0.05 in Fisher's exact tests), indicating that *Geocoris* spp. continued to explore ir*LOX2* and hemi-ir*LOX2* plants but not to predate from them over a week of infestation followed by 5 days of *M. sexta* egg predation assays (during which *M. sexta* eggs were simultaneously applied to all genotypes); and we followed predation of *M. sexta* larvae from all four genotypes in parallel over 1 week, during which ir*LOX2* and hemi-ir*LOX2* were infested with more larvae than WT and ir*PI* due to sustained higher predation rates on WT and *irPI*; but predation remained higher on WT and ir*PI*.

Predation of both larvae and eggs without GLV supplementation was two to four times as great on GLV-emitting WT and ir*PI* plants: 43%/60% (WT/ir*PI*) for larvae and 34%/39% for eggs, vs 17%/33% (ir*LOX2*/hemi-ir*LOX2*) for larvae and 9%/20% for eggs (Fisher's exact tests: N=30 larvae, p=0.047 for ir*LOX2* vs WT, p=0.069 for hemi-ir*LOX2* vs ir*PI*; N=88 eggs, p<0.001 for ir*LOX2* vs WT, p=0.013 for hemi-ir*LOX2* vs ir*PI*) (***Figure 5***). Predation was associated with a steady *Geocoris* spp. population of 16–23 individuals per day within a 5 cm radius around plants (***Table 2***). However, there was no difference among plant genotypes in the number of *Geocoris* spp. individuals (p>0.05 in Fisher's exact tests), indicating that *Geocoris* spp. regularly survey all plants and use GLVs as a short-distance cue to determine which plants harbor prey. ***Figure 5*** shows larval predation rates at the beginning of the

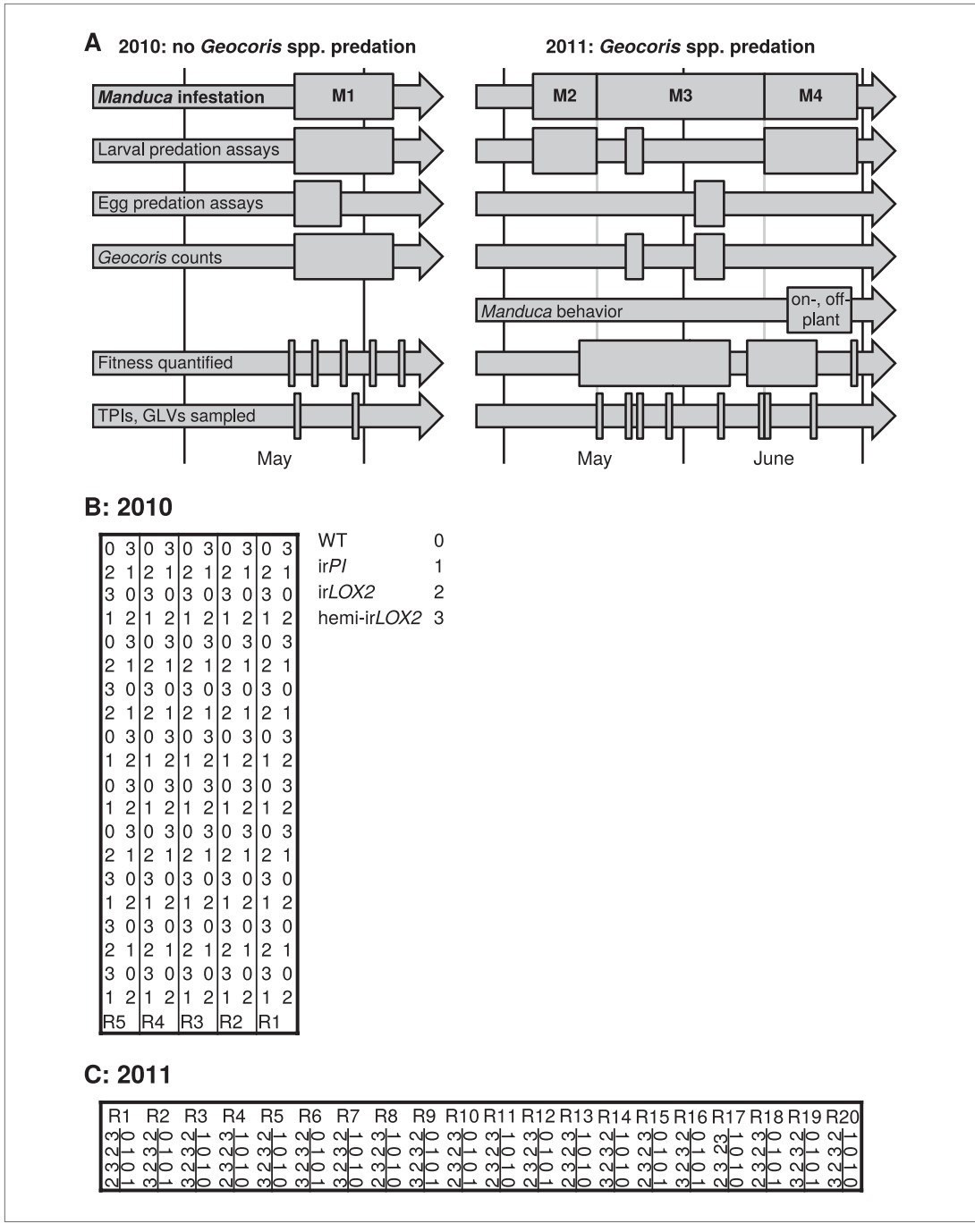

**Figure 4**. Experimental timeline and layout. (**A**) Timeline of field experiments in 2010 and 2011. Different assays and measurements are represented by individual arrows, and rectangles span the time frame of each assay or measurement; narrow rectangles represent single days. Four experimental *Manduca* infestations (M1–M4) structure the overall experimental design: M1–M3, with laboratory *Manduca*, and M4, with wild *Manduca* larvae. (**B** and **C**) Layouts of field plots in (B) 2010 and (C) 2011. Thick lines denote the borders of the experiment, thin lines denote irrigation lines (vertical borders of plot were also irrigation lines in [B] 2010), and R# denotes row number (used for identifying replicates during the experiment). The genotype key in (B) applies to both (B) and (C).

**Table 1.** GLV mix used to externally supplement plant GLV emission in M2 (see *Figure 4*) (*Allmann and Baldwin, 2010*)

| Component | Nanogram/20 µL lanolin |
| --- | --- |
| (Z)-hex-3-enal | 3530 |
| (E)-hex-2-enal | 2690 |
| (Z)-hex-3-en-1-ol | 1780 |
| (E)-hex-2-en-1-ol | 2440 |
| (Z)-hex-3-enyl acetate | 46.6 |
| (E)-hex-2-enyl acetate | 35.5 |
| (Z)-hex-3-enyl propanoate | 9.00 |
| (E)-hex-2-enyl propanoate | 8.08 |
| (Z)-hex-3-enyl butanoate | 97.0 |
| (E)-hex-2-enyl butanoate | 35.6 |

Pure GLVs were diluted in 1 mL of hexane and mixed into 14 mL of lanolin to yield the amount shown per 20 µL, representing the emission per g leaf material within the first 20 minutes of W+OS elicitation. Lanolin containing an equivalent amount of hexane was used as a control.

assay, when the *M. sexta* load was comparable across plant genotypes. Over the following week, *Geocoris* spp. predated a total of 80% of these larvae from WT and ir*PI* vs 47% from ir*LOX2* (Fisher's exact test, p=0.015 vs WT) and 67% from hemi-ir*LOX2* (p=0.382 vs ir*PI*).

In summary, *Geocoris* spp. had the same opportunity to locate *M. sexta* larvae and eggs on all genotypes, but consistently preferred to predate from GLV-supplemented or -emitting plants.

## *Manduca* spp. damage reduces plant growth and reproduction

We took the different number of 'days in field' for each plant into account in our comparison of growth and reproduction among genotypes and therefore the staggered planting did not affect this comparison (*Figure 6*, statistics *Table 3*). The ir*LOX2* and hemi-ir*LOX2* plants suffered, in total, a similar amount of *M. sexta* damage to WT plants in trials M2 and M3 (*Figure 4*), and only ir*PI* plants suffered significantly less *M. sexta* damage (*Figure 7*).

Reduced predation of *M. sexta* from ir*LOX2* and hemi-ir*LOX2* in trials M2 and M3 in 2011 (*Figure 4*) correlated with the reduced growth and reproduction of both genotypes, by 30–50% for ir*LOX2* and 20–30% for hemi-ir*LOX2* vs WT, although this reduction was also apparent in plants not infested with *M. sexta*. (*Figures 4 and 6*, statistics *Table 3*). In 2010 however, in the absence of predation, there was no difference in stem growth, branching, or bud and flower production among genotypes irrespective of *M. sexta* infestation (*Figure 6*, statistics *Table 3*). Although *M. sexta* feeding significantly affected growth and reproduction of plants overall, the effect was not significant for ir*LOX2* or ir*PI* plants in either year (*Figure 6*, statistics *Table 3*), possibly due to reduced feeding damage resulting from a lack of TPI-induced compensatory feeding in ir*PI* (*Steppuhn and Baldwin, 2007*) (Mann-Whitney U-test between ir*PI* and WT on May 28, U=54, p=0.046, *Figure 7*). Although GLVs are feeding stimulants (*Halitschke et al., 2004*), we could not measure reduced *M. sexta* feeding damage in hemi-ir*LOX2* or ir*LOX2* (*Figure 7*). Yet hemi-*irLOX2* plants, despite strongly reduced GLVs, still suffered reduced growth and reproduction from *Manduca* spp. feeding: *M. sexta* feeding reduced flower production rates by about 50% in WT and by about 30% in hemi-ir*LOX2* plants in 2010, although the overall reduction was only significant in WT; and reduced bud production significantly for both WT and hemi-ir*LOX2* by 25–30% in 2011 (*Figure 6*, statistics *Table 3*).

## Damage from naturally occurring herbivores other than *Manduca* spp. cannot explain differences in plant fitness

We monitored herbivore attack to determine whether GLV-silenced plants suffered different amounts of damage from naturally occurring herbivores, which could also cause differences in their growth and reproduction. All genotypes were attacked by mirid (*Tupiocoris notatus*) and noctuid herbivores

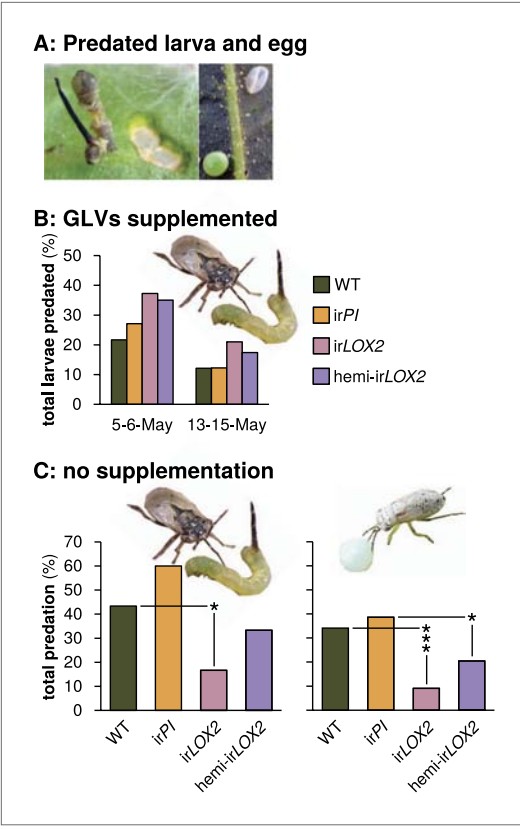

**Figure 5**. Predation of *M. sexta* larvae and eggs by *Geocoris* spp. (**A**) Examples of predated *M. sexta* larva (left panel) and egg (right panel). Left, the carcass of a predated first-instar *M. sexta* larva and typical feeding damage from early-instar *Manduca* spp. larvae. Right, an intact (lower left) and a predated (upper right) *Manduca* spp. egg. In this case, the predated egg collapsed during predation. (**B**) Total predation of *M. sexta* larvae per trial over two trials during infestation M2. GLVs were supplemented externally by placing cotton swabs next to *Manduca*-infested leaves (1 per plant). Cotton swabs next to ir*LOX2* and hemi-ir*LOX2* plants received 20 μL of a GLV mixture in lanolin paste (*Table 1*); those next to WT and ir*PI* plants received lanolin with hexane as a control because hexane was used to dissolve GLVs before mixing with lanolin. N=59–60 larvae on May 5–6 and 92–100 larvae on May 13–15. *Geocoris* spp. tended to predate more larvae from GLV-supplemented plants (Fisher's exact tests, 35–37% vs 22–27% May 5–6, p=0.066; 17–21% vs 12% May 13–15, p=0.069; combined trials, Bonferroni-corrected p=0.0063). (**C**) Total percentage of *M. sexta* larvae (left panel, N=30 larvae) and eggs (right panel, N=88 eggs) predated in two separate trials during infestation M3 in 2011 (*Figure 4*). There was no predation of larvae or eggs by *Geocoris* spp. in 2010. Raw data for (B) and (C) is in F5BC_SchumanBarthelBaldwin2012predation. xlsx (Dryad: ***Schuman et al., 2012***). Pictures are of a *G. pallens* adult predating a first-instar *Manduca* spp. larva (left) and a fifth-instar *G. pallens* nymph predating a *Manduca* spp. egg (right picture, S. Allmann). *p<0.05, ***p<0.001 in Fisher's exact tests against WT (ir*LOX2*) or ir*PI* (hemi-ir*LOX2*, which also contains the ir*PI* construct).

which caused similar amounts of damage across genotypes and years (ca. 15% and 3% of total canopy area, respectively) although ir*LOX2* plants suffered 60% less mirid and noctuid damage by the end of M3 in 2011 (N=24–28; Bonferroni-corrected Kruskal-Wallis test, noctuids May 5, p=0.027, all pairwise tests Bonferroni-corrected p>0.05; one-way ANOVAs with factor genotype on arcsine-transformed data: mirids May 27, $F_{3,103}$=5.291, p=0.002, p<0.05 for ir*LOX2* vs hemi-ir*LOX2* and ir*PI* in Scheffe *post hoc* tests, noctuids May 27, $F_{3,103}$=3.503, p=0.018, all *post hoc* tests p>0.05; all other comparisons p>0.05; *Figures 4 and 7*). Reduced herbivore damage on ir*LOX2* in 2011 could have *increased* the growth and reproduction of ir*LOX2* plants relative to WT, but cannot explain why ir*LOX2* plants instead displayed *reduced* growth and reproduction. Plants in 2011 were also damaged by flea beetles and grasshoppers (<3% of canopy area, Kruskal-Wallis tests, N=24–28, all comparisons p>0.05, *Figure 7*).

We cannot exclude the possibility that reduced growth and reproduction of uninfested ir*LOX2* and hemi-ir*LOX2* plants in 2011 (*Figure 6*, statistics *Table 3*) might have been due to non-herbivory-related factors (e.g., differences in root health corresponding to GLV antimicrobial properties) which did not play a role in 2010. Because of this uncertainty, we conducted assay M4 (*Figure 4*) in which plants were carefully matched for size and prior reproduction (*Figure 8*), and this experiment is the more robust basis for our argument that GLV-mediated indirect defense increases plant reproduction.

## GLV-mediated *Manduca* spp. mortality positively correlates to plant reproduction

To ensure that the correlated differences we observed in plant reproduction and *M. sexta* mortality were due to plant GLV emission and not to different timing and amounts of *M. sexta* damage, and to avoid the influence of non-herbivory-related factors, we conducted a *Manduca* spp. predation and plant performance assay during infestations M1 in 2010, and M4 in 2011 (*Figure 4*) for which all plants used were matched for size, as well as former damage and reproduction as necessary (*Figure 8*), and infested simultaneously with *Manduca* spp. neonates. We hypothesized that the 50% lower predation rates of *Manduca* spp. from GLV-deficient plants (*Figure 5*), combined with *Manduca* spp.'s negative effect on growth and reproduction (*Figure 6*), would result in reduced reproduction for

**Table 2.** Numbers (N) of *Geocoris* spp. individuals (nymphs and adults) within 5 cm radii around plants used for predation experiments, counted within half an hour during the main period of *Geocoris* spp. activity.

| Experiment | Genotype | *Geocoris* spp. per day (n) | | | | | Plants (n) |
|---|---|---|---|---|---|---|---|
| *Larval predation* | *Dates* | May | 21 | 22 | | | |
| May 21–23, 2011 | WT | | 3 | 4 | | | 19 |
| | ir*PI* | | 6 | 6 | | | 24 |
| | ir*LOX2* | | 6 | 4 | | | 20 |
| | hemi-ir*LOX2* | | 8 | 2 | | | 20 |
| | Total | | **23** | **16** | | | **83** |
| *Egg predation* | Dates | June | 3 | 4 | 5 | 7 | |
| June 2–6, 2011 | WT | | 2 | 5 | 2 | 1 | 18 |
| | ir*PI* | | 3 | 7 | 1 | 5 | 21 |
| | ir*LOX2* | | 4 | 2 | 0 | 3 | 21 |
| | hemi-ir*LOX2* | | 1 | 1 | 2 | 2 | 24 |
| | Total | | **10** | **15** | **5** | **11** | **84** |

Numbers are shown as subtotals for each plant genotype and grand totals per day (in bold).

GLV-deficient versus matched GLV-producing plants if *Geocoris* spp. were present. Homozygous ir*LOX2* plants were excluded from these 'matched' experiments because they did not suffer reduced growth or reproduction from *M. sexta* feeding, and because they were too small in comparison to other lines in 2011 (*Figure 6*).

In both 2010 and 2011, we selected triplets of WT, ir*PI* and hemi-ir*LOX2* plants similar in size, reproductive output, apparent health, and prior damage; damage from naturally occurring herbivores did not differ among these genotypes (*Figure 7*). In 2010, matched plants were part of infestation M1 (*Figure 4*) and thus it was not necessary to control for prior reproduction or *M. sexta* damage. Plants in 2010 received three lab strain *M. sexta* larvae per plant to a lower stem leaf. We recorded the mortality of *M. sexta* larvae and the reproductive output of plants until they began to set unripe seed. No reproductive meristems were removed, but flowers were removed and counted periodically over the first 10 days, as was done during infestation M3 in 2011 (*Figures 4 and 6*, statistics *Table 3*), to track plant reproduction while avoiding ripe seed capsules: the distribution of ripe seed is not permitted for genetically modified plants. In the absence of *Geocoris* spp. in 2010, genotypes did not differ in *M. sexta* mortality (N=51 larvae)—which in every observed case was due to a failure of the larva to feed—or plant reproduction (N=17 plants) (*Figure 9*). This, and the fact that flower production did not differ among genotypes in 2011 through infestation M3 despite flower removal (*Figure 6*), indicates that flower removal itself does not cause a difference among genotypes, and suggests that the other differences among genotypes in growth and reproduction seen in 2011 (*Figure 6*, statistics *Table 3*) are real.

In 2011, hemi-ir*LOX2*, ir*PI* and WT plants were matched prior to infestation M4 to exclude differences in growth, reproduction and *Manduca* spp. damage arising during *M. sexta* infestations M2 and M3 and from caged *Manduca* spp. during the egg predation assay (*Figures 4, 7, and 8*). Instead of regularly removing flowers, we removed all reproductive meristems from matched plants in 2011 by cutting inflorescences at their base. This allowed us to follow a new set of reproductive meristems through to seed set without incurring ripe seed. Because plants were matched prior to the assay, a similar number of reproductive meristems were cut from all plants, and thus all plants were similarly affected by this cutting (*Figure 8*, see 'Discussion').

Because oviposition by native *Manduca* spp. moths provided sufficient eggs prior to the beginning of M4 (*Figure 4*), we decided to conduct this infestation with wild larvae and thereby demonstrate that native larvae, like larvae of the lab strain, are susceptible to GLV-mediated predation. To make M4 a realistic test, we placed one wild *Manduca* spp. neonate per plant on a lower stem leaf to mimic natural oviposition rates (*Kessler and Baldwin, 2001*). We again recorded the mortality of *Manduca* spp. larvae and the new reproductive output of plants until they began to set unripe seed. During the first

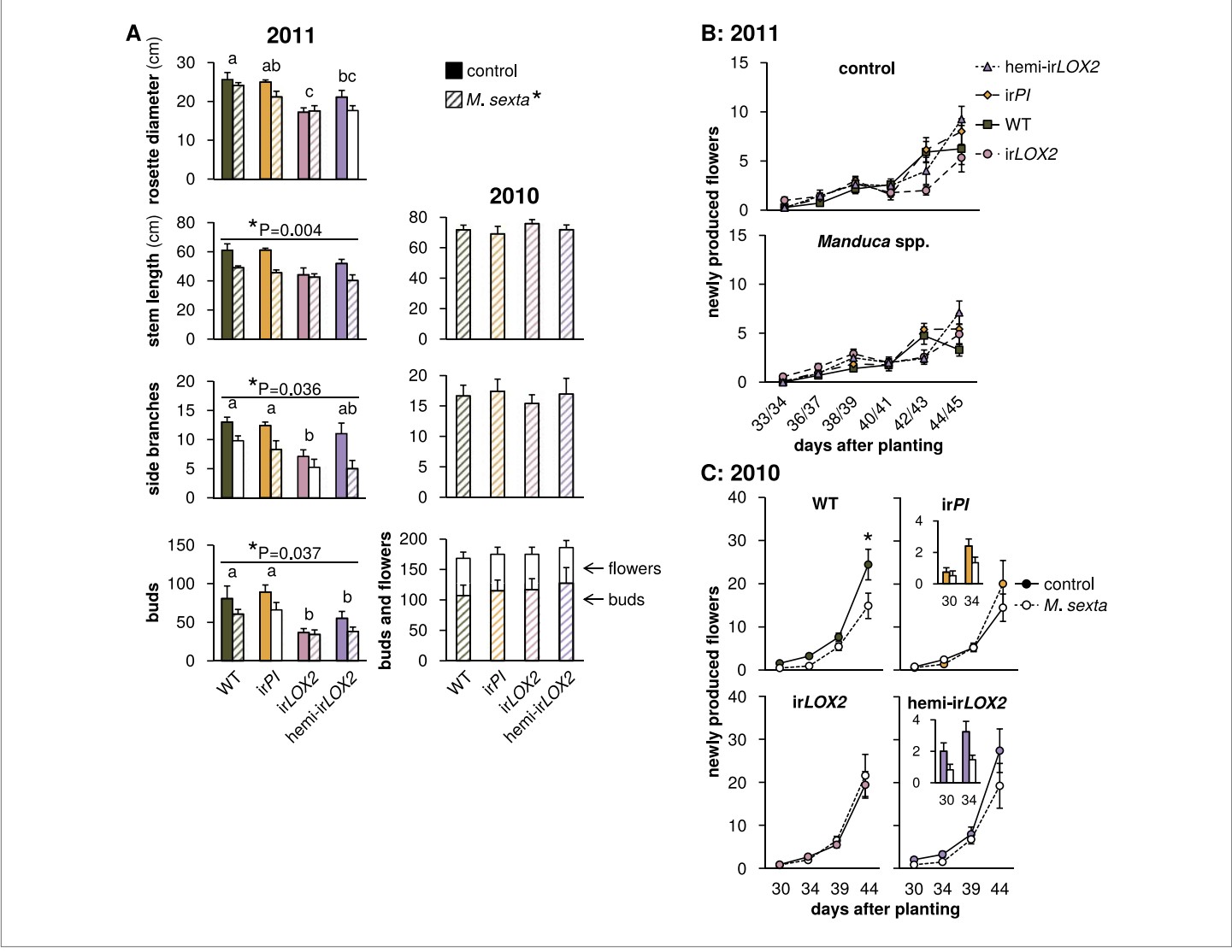

**Figure 6**. Growth and reproduction of plants during the 2010 and 2011 field seasons; graphs show means±SEM. (**A**) Final growth measurements for *M. sexta*-infested and uninfested control plants of each genotype in 2011 (left, 44–45 days after planting, N=11–17) or *M. sexta*-infested plants in 2010 (right, June 6, 2 days after the removal of fifth-instar *M. sexta* larvae). *p<0.05 for Wilks' Lambda test of the effect of *M. sexta* feeding on growth and reproduction in 2011, day 44–45, in a two-way MANOVA with factors genotype and treatment ($F_{6,52}$=2.287, p=0.049). *p-values above individual graphs denote the significance of *M. sexta* feeding over all genotypes in 2011 for the measurement shown in the MANOVA, or in a separate Mann-Whitney U-test for side branches (stem $F_{1,57}$=9.155; side branches, U = 270; buds $F_{1,57}$=4.572); values for individual genotypes are in **Table 3**. [a, b, c] Different letters denote significant (p<0.05) differences between genotypes in 2011 for Scheffe *post hoc* tests (rosette diameter $F_{3,57}$=8.791, p<0.001, stem length $F_{3,57}$=4.192, p=0.009, number of buds $F_{3,57}$=9.876, p<0.001) or Bonferroni-corrected p-values for Mann-Whitney U-tests following a Kruskal-Wallis test (side branches $\chi^2$ = 10.958). In 2010, in the absence of *Geocoris* spp. activity, there were no significant differences between genotypes in the parameters shown with or without *M. sexta* infestation (**Table 3**). Bud numbers from 2010 are also shown in Figure 9. (**B** and **C**) Flower production for *M. sexta*-infested and uninfested control plants from the beginning of flowering in (B) 2011 and (C) 2010. Flowers were counted and removed at the time points shown: each time point represents new flower production. Insets in (C) show the first two time points for ir*PI* and hemi-ir*LOX2*. *p<0.05 for the main effect of *M. sexta* infestation in a repeated-measures ANOVA with log$_2$-transformed data (**Table 3**). Raw data for 2011 is in F6AB_SchumanBarthelBaldwin2012growth_reproduction2011.xlsx and T4_SchumanBarthelBaldwin2012growth_reproduction2011.xlsx, and data for 2010 is in F6AC_SchumanBarthelBaldwin2012growth_reproduction2010.xlsx (Dryad: **Schuman et al., 2012**).

**Table 3.** Results of Mann–Whitney U-tests, Kruskal–Wallis tests, and ANOVAs for control vs *M. sexta*-infested plants of each genotype grown in the field in 2010 and 2011 (**Figures 6 and 9**)

| 2010 | | Branches | | | Stem, buds, flowers | | |
|---|---|---|---|---|---|---|---|
| | | Mann–Whitney, Kruskal–Wallis | | | MANOVA, Wilks' lambda | | |
| Comparison | Genotype | df | $\chi^2$ | p* | df | F | p |
| Treatment | All | 1 | 0.022 | 1.000 | 3, 148 | 0.463 | 0.709 |
| Genotype | All | 3 | 2.909 | 0.802 | 9, 360.344 | 1.186 | 0.303 |

| 2011 | | Branches (n) | | | Rosette diameter (cm) | | | Stem length (cm) | | | Buds (n) | | | Flowers (n) | | |
|---|---|---|---|---|---|---|---|---|---|---|---|---|---|---|---|---|
| | | Student's t-test | | | Student's t-test | | | MANOVA, Wilks' lambda | | | MANOVA, Wilks' lambda | | | MANOVA, Wilks' lambda | | |
| Comparison | Genotype | df | t | p | df | t | p | df | F | p | df | F | p | df | F | p |
| Treatment×time | WT | 26 | 1.696 | 0.102 | 26 | −0.870 | 0.932 | 5, 22 | 3.871 | **0.011** | 5, 22 | 3.188 | **0.026** | 3, 24 | 1.213 | 0.326 |
| | ir*PI* | 26 | 1.024 | 0.315 | 26 | −0.161 | 0.873 | 5, 22 | 0.991 | 0.446 | 5, 22 | 0.656 | 0.660 | 5, 22 | 0.525 | 0.755 |
| | ir*LOX2* | 25 | 1.112 | 0.277 | 25 | −0.058 | 0.954 | 5, 21 | 0.606 | 0.696 | 5, 21 | 0.535 | 0.748 | 5, 21 | 0.540 | 0.744 |
| | hemi-ir*LOX2* | 22 | 1.753 | 0.094 | 22 | 1.140 | 0.267 | 5, 18 | 1.118 | 0.386 | 5, 18 | 3.001 | **0.038** | 4, 19 | 0.723 | 0.587 |

2010: Numbers of side branches (Mann–Whitney, Kruskal–Wallis), stem length, and final numbers of buds and flowers (MANOVA) were recorded in a single measurement at the end of M1 (**Figure 4**). Numbers of newly produced flowers were counted repeatedly upon flower removal, and Wilks' Lambda F values for the main effect of *M. sexta* feeding are shown from repeated-measures ANOVAs across all measurements; Wilks' F values for the *M. sexta*-by-time interaction were not significant. *Bonferroni-corrected p-values.

2011: Because many plants had few or no side branches before the final measurement, and rosette diameters did not change over the period that plants were measured, t-tests are shown for the final measurement of these parameters in M3 (**Figure 4**). For stem lengths, numbers (n) of buds, and numbers of flowers, Wilks' lambda F values for the *M. sexta*-by-time interaction are shown from repeated-measures ANOVAs across all measurements. Significant p-values are given in bold.

to third larval instars in which larvae are vulnerable to *Geocoris* spp. predation (**Kessler and Baldwin, 2001**), wild *Manduca* spp. mortality was 38% on hemi-ir*LOX2* plants vs 62–76% on matched WT and ir*PI* plants; the overall mortality of larvae on all three lines was significantly different (N=21 larvae, Bonferroni-corrected pairwise comparisons by Friedman tests, p<0.01) (**Figure 9**). Although *Manduca* spp. mortality on hemi-ir*LOX2* jumped to 70% in the fourth and semi-final larval instar, this was likely due to predation by whiptail lizards (*Cnemidophorus* spp.) which were present on the field plot: these lizards predate late-instar *Manduca* and are attracted to short-chain fatty-acid volatiles produced by the larvae due to ingestion of acyl sugars in plant trichomes (**Stork et al., 2011**; **Weinhold and Baldwin, 2011**).

The plants used in M4 had not previously differed in their reproduction except that hemi-ir*LOX2* plants had produced more flowers than WT, but not ir*PI* plants (**Figure 8**). By the end of the assay, the hemi-ir*LOX2* plants had produced 40–50% fewer buds and flowers than matched WT and ir*PI* plants (N=21 plants, p<0.05 in Scheffe *post hoc* tests for hemi-ir*LOX2* vs WT and ir*PI* flowers and buds following a repeated-measures MANOVA over all flower and bud counts, Wilks' Lambda for the interaction of line and day: $F_{12,110}=2.835$, p=0.002) (**Figure 9**). This reduced bud and flower production was not due to accelerated seed set: unripe seed capsules on hemi-ir*LOX2* plants were also reduced by 50% (N=21 plants, p=0.021 for hemi-ir*LOX2* vs ir*PI* in a Scheffe *post hoc* test following an ANOVA with genotype as the factor, $F_{2,60}=4.142$, p=0.021) (**Figure 9**). These data demonstrate that herbivore-induced GLV emissions function as indirect defenses by increasing predation of *Manduca* spp. larvae twofold, resulting in a twofold increase in bud and flower production for *N. attenuata* in its native habitat.

### *M. sexta* and *M. quinquemaculata* perform similarly on plants

To ensure that our results were not biased by the use of wild *Manduca* spp. larvae, which comprised both *M. sexta* and *M. quinquemaculata*, we analyzed the growth (length over time) and instar change of larvae on plants in M4 by larval species. *M. sexta* and *M. quinquemaculata* did not differ in their growth or instar progression (N=11–13, repeated measures ANOVA for days 4–11, Wilks' Lambda for

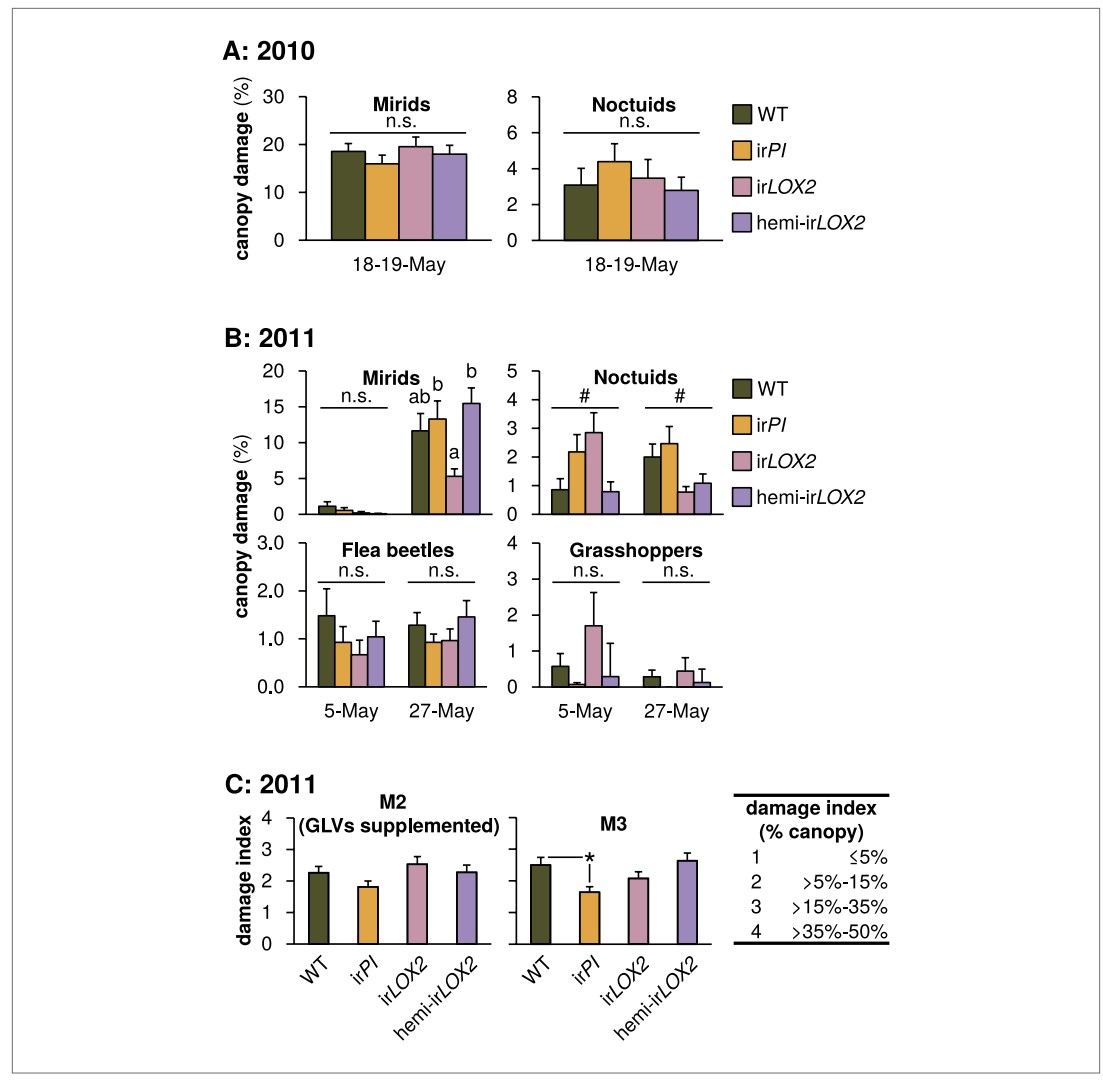

**Figure 7**. Herbivore damage to plants during the 2010 and 2011 field seasons (means+SEM). For a timeline of *Manduca* infestations M1–M4, see **Figure 4A**. (**A**) Total canopy damage due to naturally occurring herbivores before the start of infestation M1 in 2010, N=17. For raw data, see F7A_SchumanBarthelBaldwin2012herbivoreDamage2010. xlsx (Dryad: **Schuman et al., 2012**). (B) Total canopy damage due to naturally occurring herbivores before infestation M2 (May 5) and near the end of M3 (May 27) in 2011, N=24–28. [a, b] Different letters denote significant (p<0.05) differences between genotypes in Scheffe *post hoc* tests following one-way ANOVAs for arcsine-trans-formed data at each timepoint (mirids May 27 $F_{3,103}$=5.291, p=0.002; noctuids May 27 $F_{3,103}$=3.503, p=0.018); n.s.: not significantly different. #p<0.05 for the main effect of genotype on noctuid damage in a Bonferroni-corrected Kruskal-Wallis test, May 5 ($\chi^2$=11.239, p=0.027). (**C**) Damage in 2011 from *M. sexta* larvae used in the predation assays in M2 (left panel) and M3 (right panel). GLVs were externally supplemented to plants in infestation M2 and not in M3. Total canopy damage was estimated, using the index, by an independent observer without knowledge of plant identity (N=11–17). *p<0.05 in a Mann-Whitney U-test between ir*PI* and WT on May 28 (U=54, p=0.046); the difference on May 15 was not significant (p>0.1). Note that scales differ. Raw data for (B) and (C) is in F7BC_SchumanBarthelBaldwin2012herbivoreDamage2011.xlsx (Dryad: **Schuman et al., 2012**).

the interaction of day and species: $F_{11,12}$=1.356, p=0.311). Because larvae of the two species cannot be distinguished before the third instar, we could not test whether mortality was equal for both species in the first three instars; however, because other collections of wild eggs around the same time as the collection for our experiment yielded a 1:1 ratio of species, and because our ratio of the species remained 1:1 after larvae reached the third instar, it is likely that mortality of the two species was equal prior to the third instar.

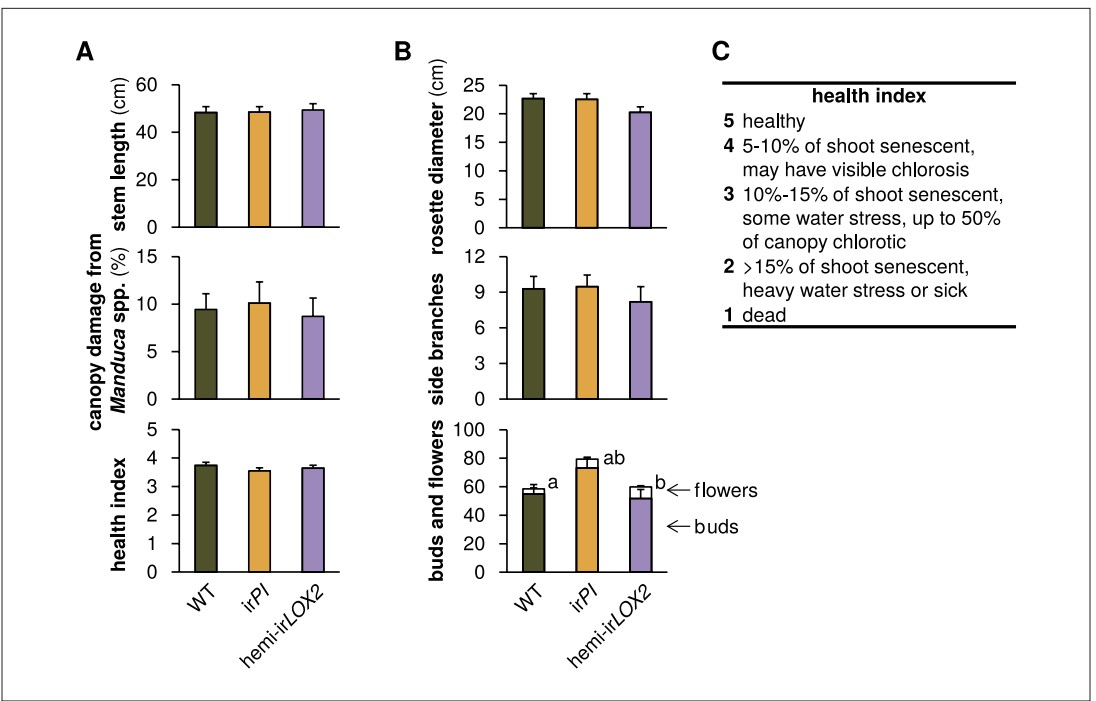

**Figure 8**. Comparison of plants used in triplets for infestation M4 in 2011 (see **Figure 4A**); graphs show means+SEM (N=21 plants). (**A**) Parameters used to match plants in triplets. Measurements and assessments are from the first day of M4. (**B**) Final measurement of prior growth and reproduction for plants used for triplets; data are from the final two measurements during infestation M3 (see **Figure 4A**). [a, b]Different letters denote significant differences (p<0.001) for flower number in Scheffe *post hoc* tests following a MANOVA with all measurements and genotype as the factor ($F_{2,60}$=8.668, p<0.001). (**C**) Health index used in (A). For raw data, see F8_SchumanBarthelBaldwin2012triplets.xlsx (Dryad: **Schuman et al., 2012**).

## *Manduca* spp. response to mock predator attack is altered by TPI consumption

TPIs had a less consistent and, contrary to our expectations, negative effect on *Manduca* spp. predation (**Figure 5**); furthermore, there was no positive effect of TPIs on plant growth and reproduction (**Figure 6**, statistics **Table 3**) and only a marginal effect of TPIs on *Manduca* spp. growth under natural conditions (N=13–26 second instar larvae during M2, one-way ANOVAs with genotype as the factor, $F_{3,77}$=2.792, p=0.046, all *post hoc* tests p>0.05, **Figure 10**; N=8 second instar larvae during M4, paired t-test between matched WT and ir*PI*, p=0.052). We hypothesized that the reduced access to protein for larvae feeding on TPI-producing plants might nevertheless affect *Manduca* spp. behavior independently of larval size. Indeed, wild *Manduca* spp. larvae feeding on WT plants (infestation M4, **Figure 4**) reacted more sluggishly to experimental provocation than size-matched larvae on ir*PI* plants: they were 75% less likely to attack when lifted off of the leaf (N=5 second-instar larvae matched for size, p=0.035 in paired t-test) (**Figure 10**, **Videos 1 and 2**).

We were careful not to harm wild larvae so that we could monitor their natural mortality and consequences for plant reproduction (**Figure 9**). To more accurately imitate *Geocoris* spp. attack, we developed an off-plant assay with larvae from the laboratory *M. sexta* strain feeding on detached leaves from field-grown plants, in which size-matched larvae were poked, pierced and lifted using an insect pin to mimic the *Geocoris* spp. beak (**Figure 10**, **Videos 3 and 4**). Similarly to the on-plant assay, larvae fed on WT leaves were 50% less likely to successfully attack the insect pin, either when initially poked, or poked and lifted with the pin (N=20 first-instar larvae matched for size, p<0.05 in paired t-tests) (**Figure 10**). We also monitored recovery post-trial and found that WT-fed larvae ceased to grow for at least 24 hr after simulated attack, while ir*PI*-fed larvae

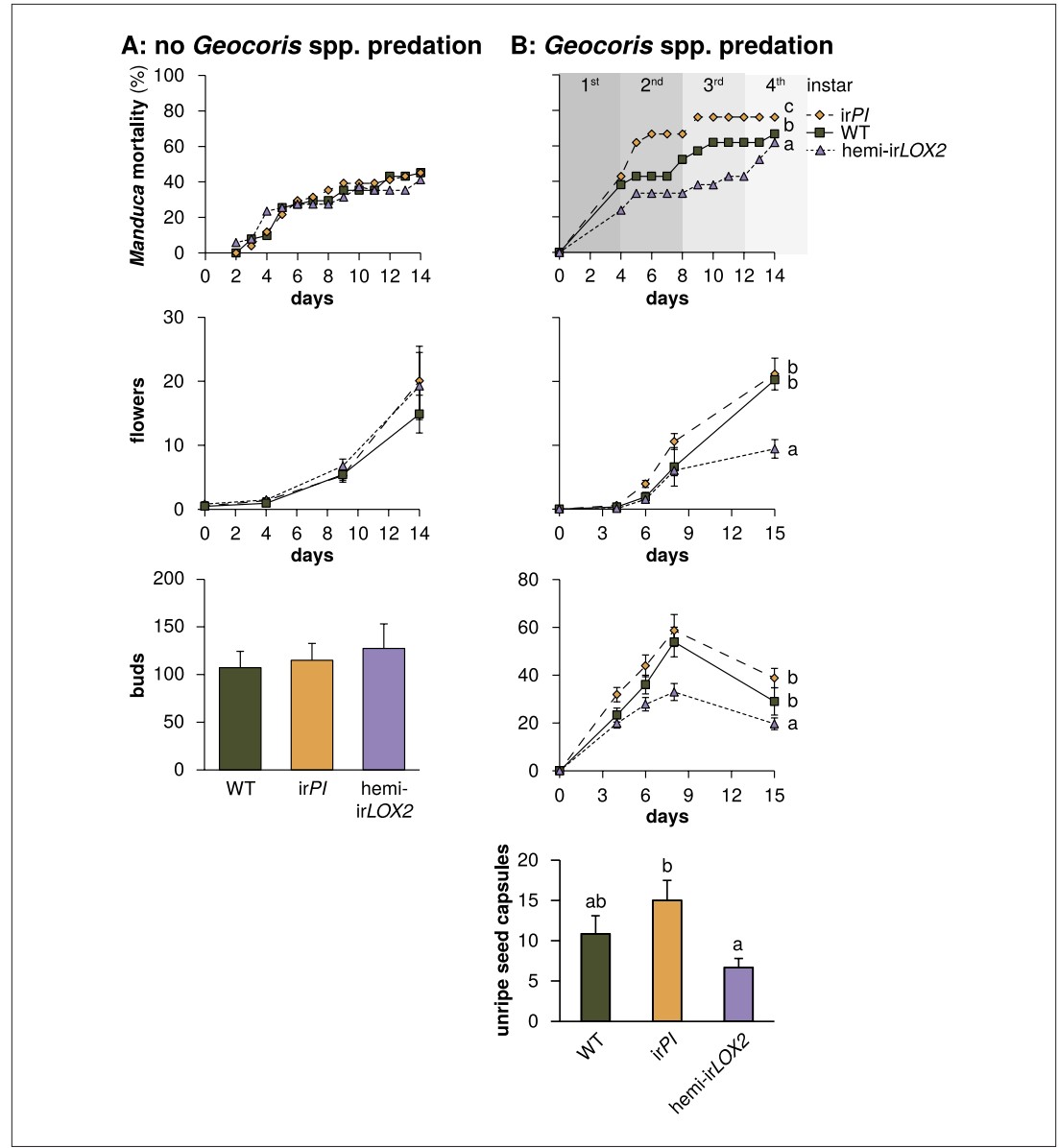

**Figure 9**. Cumulative mortality of *Manduca* spp. larvae and numbers of reproductive units produced by infested plants in 2010, in the absence of *Geocoris* spp. predation, and in 2011, when *Geocoris* spp. were active predators of *Manduca* spp. (**A**) In 2010, flowering plants matched for size (N=17) were each infested with three *M. sexta* neonates from a laboratory culture (N=51 larvae), which were allowed to reach the final instar on plants. The upper panel shows larva mortality over time, which reached a maximum of 40% by the fifth instar, after 12 days. Flower production (lower panel) did not differ, nor did any other parameters of plant size and reproduction (**Figure 6**, **Table 3**) including number of buds produced by June 6, which was day 19 after infestation and day 49 after planting in the field. For raw data, see F9A_SchumanBarthelBaldwin2012data2010.xlsx (Dryad: **Schuman et al., 2012**). (**B**) In 2011, plants (N=21) were matched for size, prior reproduction, health, and previous damage by *Manduca* spp. and other herbivores (**Figures 7 and 8**) following the end of infestation M3 (**Figure 4**), and reproductive meristems were removed. Matched plants were infested with one wild *Manduca* spp. neonate each (M4 in **Figure 4**), and *Manduca* spp. larvae were allowed to reach the fourth (penultimate) instar. Larval mortality (upper panel) reached a maximum of 76% after larvae transitioned from the second to third instar (days 9 and 10), at which time larval mortality on hemi-ir*LOX2* was only half as great as on WT or ir*PI*; larvae beyond this stage are not susceptible to *Geocoris* spp. (**Kessler and Baldwin, 2001**). Flower and bud production (lower panel) was twice as great in WT and ir*PI* as in hemi-ir*LOX2*, and numbers of flowers and buds correspond to numbers of seed capsules: hemi-ir*LOX2* plants also produced fewer unripe seed capsules than WT or ir*PI* plants. For raw data, see F9B_SchumanBarthelBaldwin2012data2011.xlsx (Dryad: **Schuman et al., 2012**). [a, b, c] Different letters indicate
*Figure 9. Continued on next page*

*Figure 9. Continued*

significant differences (p<0.01) in Bonferroni-corrected pairwise Friedman tests (*Manduca* spp. mortality), or Scheffe *post hoc* tests of hemi-ir*LOX2* versus WT and ir*PI* flowers and buds following a repeated-measures MANOVA over all flower and bud counts shown (results of Greenhouse-Geisser-corrected univariate tests for the interaction of line and day: buds, $F_{4.988,149.653}$=5.297, p<0.001; flowers, $F_{3.722,111.657}$=4.403, p=0.003), or significant differences (p<0.05) in Scheffe *post hoc* tests following an ANOVA for unripe seed capsules at day 15 with genotype as the factor ($F_{2,60}$=4.142, P=0.021).

continued to grow (p=0.016 in Student's t-test); mortality did not differ (p=0.527 in Fisher's exact test) (*Figure 10*).

Thus TPIs did not increase plant reproduction under attack from *Manduca* spp. in nature, but may support indirect defense by weakening the response of larvae to predator attack. The contradictory higher predation rates of *Manduca* spp. larvae from ir*PI* than from WT plants (*Figure 5*) might reflect *Geocoris* spp.'s feeding preference, if ir*PI*-fed larvae are more nutritious than WT-fed larvae (*Kaplan and Thaler, 2011*).

## Discussion

Our data demonstrate that herbivore-induced GLV emissions function as indirect defenses by increasing predation of *Manduca* spp. twofold, resulting in a twofold increase in bud and flower production for *N. attenuata* in its native habitat. In contrast, there was no positive effect of TPIs on plant growth and reproduction and no significant effect of TPIs on *Manduca* spp. growth under natural conditions; however, TPIs may support indirect defense by weakening the response of larvae to predator attack. Although this indicates that predation rates from ir*PI* plants should be reduced, we observed a tendency towards higher predation rates from ir*PI* than from WT plants; this could reflect *Geocoris* spp.'s preference if ir*PI*-fed larvae are more nutritious than WT-fed larvae (*Kaplan and Thaler, 2011*).

### WT levels of TPIs in hemi-irLOX2 plants are likely due to gene dosage effects

The hemi-ir*LOX2* plants used in this study were created by crossing homozygous ir*PI* and ir*LOX2* plants, but the ir*PI* construct was not active in the hemizygous state (*Figure 1*). We continued to use this cross for its less severely reduced GLV production in comparison to homozygous ir*LOX2* plants (*Figures 2 and 3*), which likely permitted growth and reproduction comparable to ir*PI* and WT in 2011 that was essential for plant matching prior to the final assays of *Manduca* spp. mortality and plant reproduction (M4, *Figures 4 and 9*). It is common molecular biology knowledge that functional RNAi constructs may be rendered ineffective as a result of insufficient gene dosage, for example, *Travella et al. (2006)* and references therein, which may occur when an RNAi construct is present in the hemizygous state (*García-Pérez et al., 2004*). The 35S promoter which drives the transcription of the RNAi construct may also be methylated: an epigenetic effect which can reduce the dose of RNAi in individual plants within a single transformed line (A Weinhold, unpublished data). This may have occurred in the ir*PI* parent used for the creation of the hemi-ir*LOX2* line, although loss of activity of the ir*PI* construct was not observed in the homozygous ir*PI* line over the lifetime of plants in the field (*Figure 1*).

### Our best measures of reproduction for transgenic plants in the field demonstrate the positive effect of GLV-mediated indirect defense

Although the production of viable offspring is the accepted definition of Darwinian fitness, we are not permitted to allow transgenic plants to disperse ripe seed in the field, and measures to prevent seed dispersal, such as bagging meristems, strongly affect production of buds and flowers and can also affect seed viability by increasing temperature, and decreasing respiration and photosynthesis of reproductive tissue and associated green tissue.

For field-grown *N. attenuata* plants, fewer than 5% of buds and flowers (in total) are aborted by healthy (not diseased) plants, and abortion seems always to be due to damage by insects (M C Schuman and I T Baldwin, personal observation, June 2010). Plants are self-compatible and more than 70% of seed set from plants in native populations results from fertilization via self-pollen

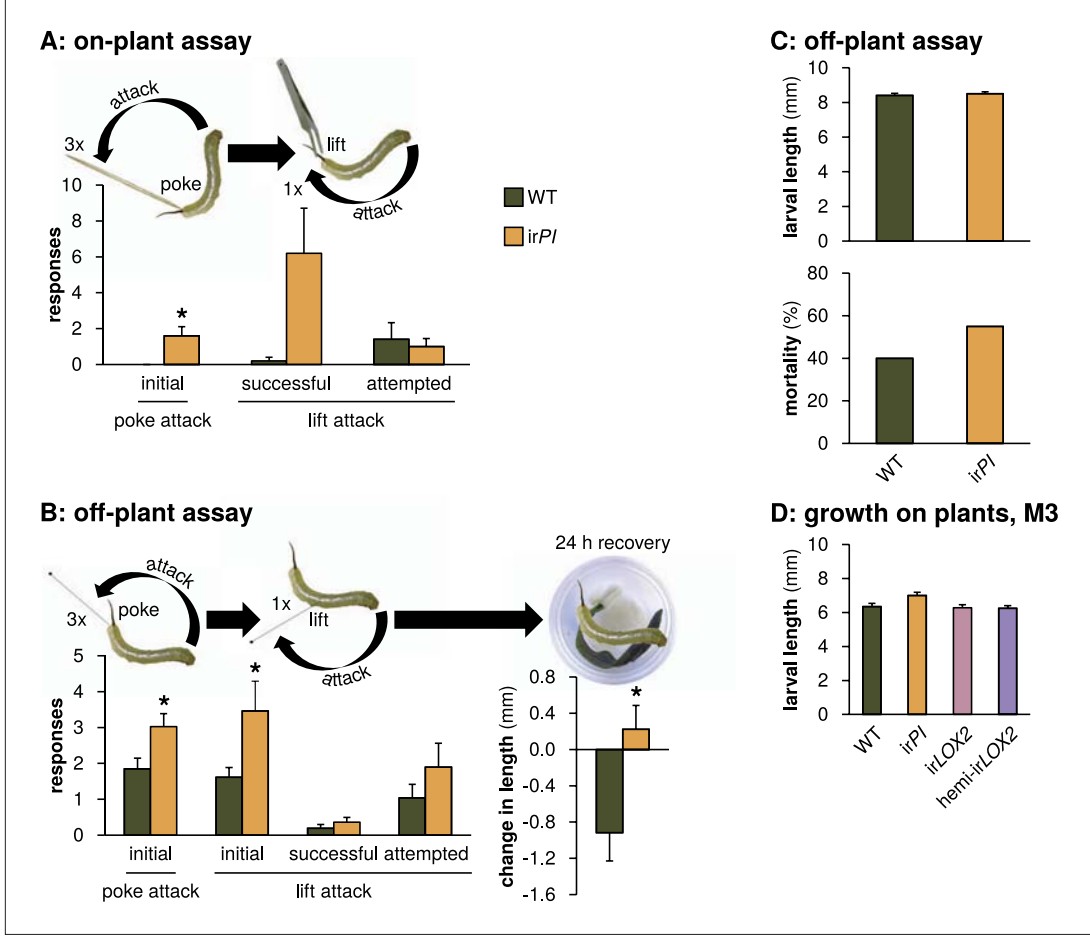

**Figure 10**. Mock *Geocoris* spp. predation assays with *Manduca* spp. larvae fed on WT or ir*PI* plants. (**A**) Response of wild *Manduca* spp. (***Figure 4A***) on plants in the field to poking with a toothpick and lifting with a featherweight forceps (N=5 second-instar larvae matched for size). We first poked larvae below the horn three times, 3 s apart, with the end of a toothpick and counted how often they attacked the toothpick, defined as the larva whipping its head around toward the toothpick and making contact. We then lifted larvae from the plant using the forceps and counted how often they attempted to attack, or succeeded in attacking the forceps over 15 s. In an attempted attack, the larvae moved from hanging at a 180° angle below the forceps vertically toward the forceps; and in a successful attack, the front end of the larva made contact with the forceps, before returning to its original position. All individuals were recorded and responses were counted from videos (see ***Videos 1 and 2***). *p<0.05 in a paired t-test. (**B**) Left, response of *M. sexta* from a laboratory strain raised for 48 hr in boxes on either WT or ir*PI* leaf tissue (N=20 first-instar larvae matched for size) to being poked, pierced and lifted with an insect pin. Right, growth of larvae in the following 24 hr. The procedure was identical to that for the on-plant assay described above, except that larvae were poked with an insect pin rather than a toothpick, and then pierced in the rear flank and lifted with the same insect pin (see ***Videos 3 and 4***). *p<0.05 in a paired t-test. The length of each larva was measured prior to poking and lifting. Afterward, larvae were placed in individual cups, each with a moist paper towel round and fresh WT or ir*PI* leaf tissue, and length of the larvae in millimeters was again measured after 24 hr; mortality did not differ between WT- and ir*PI*-fed larvae. *p<0.05 in a Student's t-test. (**C**) Upper panel, length of first instar larvae fed for 2 days on WT or ir*PI* tissue and size-matched for use in the off-plant behavioral assay mimicking *Geocoris* attack (B); lower panel, mortality of first instar larvae 24 hr after mock *Geocoris* attack as described in (B). Mortality was not significantly different in a Fisher's exact test. (D) Larval length in the first instar after 2 days on plants in the field: larvae on ir*PI* were not significantly larger. Length of surviving larvae was measured in a predation assay during infestation M3 (***Figures 4 and 5C***), N=13–26 larvae. Length was not significantly different for larvae feeding on ir*PI* in a one-way ANOVA with genotype as the factor ($F_{3,77}$=2.792, p=0.046, all *post-hoc* tests p>0.05). For raw data, see F10_SchumanBarthelBaldwin2012Manduca.xlsx (Dryad: ***Schuman et al., 2012***).

(*Sime and Baldwin, 2003*). Thus numbers of buds and flowers correlate to lifetime seed capsule production, which in turn correlates to lifetime seed production, which has been used as a proxy measure of Darwinian fitness (*Baldwin, 1998*; *van Loon et al., 2000*; *Hoballah and Turlings, 2001*). The transgenic lines used do not vary in seed mass or their seedling viability under laboratory conditions.

### Meristem removal from plants before infestation M4 was necessary despite matching, and affected all plants similarly

During the 2011 experiment, we saw a large and reproducible difference in predation from GLV-emitting versus GLV-deficient plant genotypes, which was not observed in 2010 due to an absence of *Geocoris* spp. predators in that year. This difference in predation rate correlated to a difference in plant growth and reproduction which was also not observed in 2010. To rigorously test the consequences of GLV-mediated predation of *Manduca* spp. on plant reproduction, we selected triplets of WT, ir*PI* and hemi-ir*LOX2* plants similar in size, previous reproductive output, apparent health, and prior damage to carry out *Manduca* spp. mortality and plant reproduction assays (M1 in 2010, M4 in 2011, *Figure 4*). We removed all reproductive meristems from matched plants in 2011 to allow us to follow plant reproduction over full *Manduca* spp. larval development without incurring ripe transgenic seed capsules. In 2010 (and during infestation M3 in 2011, *Figure 4*), we had removed and counted flowers regularly to track reproduction while avoiding ripe seed; this did not cause a difference in reproduction among genotypes (*Figure 6*), but we elected to avoid flower removal during infestation M4 by removing reproductive meristems prior to the beginning of the assay.

The hemi-ir*LOX2* plants chosen in 2011 had produced more flowers than WT – but not more than ir*PI*—prior to the start of infestation M4 (*Figures 4 and 8*). This did not correspond to more cuts on average for hemi-ir*LOX2* when removing reproductive meristems: meristems were cut at the bases of inflorescences which contained mostly buds, and the number of these did not differ for the plants chosen, nor did the number of side branches (*Figure 8*) which bore most reproductive meristems. Therefore, in the absence of additional effects during infestation M4, the reproduction of the matched hemi-ir*LOX2* plants should have been similar to that of WT and ir*PI*.

### Conclusions and outlook

By indicating the long-sought indirect defensive function of HIPVs, these data set the stage for the use of HIPVs as part of integrated pest management strategies (IPM), which rely in part on recruiting biological control agents to reduce pesticide use (*Horne and Page, 2008*). These agents are usually naturally occurring generalist parasitoids and predators, such as *Geocoris* spp. (*Eubanks and Denno, 1999*, *2000*; *Allison and Hare, 2009*; *Allmann and Baldwin, 2010*). HIPVs are produced by genotypes of most, if not all crop plants and IPM would benefit from selective breeding or engineering of HIPV emission (*Kos et al., 2009*) rather than relying on alternatives such as controlled release dispensers, which have mixed success and require large amounts of synthetic HIPVs (*Kaplan, 2012*). PIs may be employed to enhance the efficiency of indirect defense, especially combined with toxins like Bt that directly target herbivores and are safe for biological control agents. With growing concerns about field-evolved Bt resistance (*Liu et al., 2010*), indirect defenses promise an effective 'first line of defense' against agricultural pests, to which not even specialist herbivores are likely to rapidly evolve resistance.

## Materials and methods

### Plants, growth conditions and field plantations

Seed germination, glasshouse growth conditions, and the *Agrobacterium tumefaciens* (strain LBA 4404)–mediated transformation procedure have been described previously (*Krügel et al., 2002*). Seeds of the 31st generation of the inbred 'UT' line of *Nicotiana attenuata* (Torr. ex S. Wats.) were used as the wild-type plant in all experiments. For the field experiment, seedlings were transferred to 50 mm peat pellets (Jiffy) 15 days after germination and gradually hardened to the environmental conditions of high sunlight and low relative humidity over 10 days. Small, adapted, size-matched rosette-stage plants were transplanted into a field plot in a native habitat in Utah and watered thoroughly once at planting and as needed over the first 2 weeks until roots were established; all plants received the same watering regime in each year. WT, ir*PI*, ir*LOX2* and hemi-ir*LOX2* plants were arranged in quadruplets (N=40–50) of one plant per genotype, with individuals 0.5 m

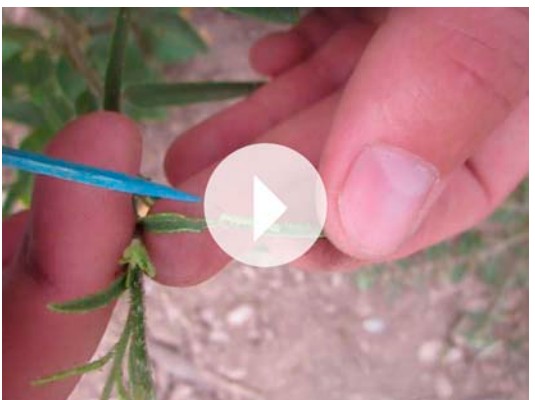

**Video 1**. On-plant assay, plant 7u, WT, June 18, 2011. DOI:10.7554/eLife.00007.016

apart, a distance sufficient to allow predators and herbivores to distinguish volatiles from neighboring plants (*Kessler and Baldwin, 2001*). Quadruplets were arranged so that no two adjacent plants were of the same genotype (*Figure 4*). In 2010, the field plot was a first-year plot located at latitude 37.141, longitude 114.027; in 2011, plants were planted at a second, older field site across a river from the first, located at latitude 37.146, longitude 114.020. Field plantations were conducted under APHIS permission numbers 06-242-3r-a3 (2010 and 2011) and 10-349-102r (2011).

We used previously characterized, homozygous, inverted-repeat (ir) RNAi transformed lines of the second transformed generation ($T_2$) to silence GLV biosynthesis: ir*LOX2* line number A-04-52-2 (*Allmann et al., 2010*), and TPI activity: ir*PI* line number A-04-186-1 (*Steppuhn and Baldwin, 2007*). Vector construction and the pSOL3 plasmid have been described previously (*Bubner et al., 2006*). A cross was created between ir*LOX2* and ir*PI* homozygous lines; however, the hemizygous ir*PI* construct did not silence TPI activity or transcripts, and these plants therefor served as vector controls for comparison with ir*PI* and had slightly greater residual GLV production than ir*LOX2* (see 'Results'). They are thus referred to as hemizygous (hemi-) ir*LOX2* plants.

### *Manduca* spp. eggs and larvae

Wild *Manduca* spp. eggs were collected for field assays when available from natural ovipositions. *M. sexta* and *M. quinquemaculata* (hereafter *Manduca* spp.) were both ovipositing at the time experiments were conducted; the species of larvae was identified at the third instar and recorded (earlier instars of these two species cannot be distinguished morphologically). *M. sexta* and *M. quinquemaculata* oral secretions (OS) are highly similar in their composition (*Halitschke et al., 2001*) and elicit similar volatiles (*Halitschke et al., 2001*; *Kessler and Baldwin, 2001*) and defense genes (*Schittko et al., 2001*) in *N. attenuata*. Eggs from laboratory-reared *M. sexta*, kindly provided by Dr. Carol Miles at SUNY Binghampton, were used in the field when wild *Manduca* spp. eggs were not sufficiently abundant. Eggs were allowed to hatch in well-aerated boxes on fresh *N. attenuata* leaf tissue over a moistened paper towel. *M. sexta* larvae used to elicit glasshouse-grown plants, or to collect oral secretions (OS) for plant treatments, were taken from an in-house colony at the Max Planck Institute for Chemical Ecology in Jena, Germany.

### *Manduca* spp. infestation and W+OS treatment of plants

Because GLVs influence *Manduca* spp. oviposition (*De Moraes et al., 2001*; *Kessler and Baldwin, 2001*; *Fraser et al., 2003*), and the timing and extent of *Manduca* spp. oviposition varies from year to year, we created even, synchronous oviposition events by infesting plants with *Manduca* spp. larvae, either from a lab-reared culture or from wild collections (see Manduca *spp. eggs and larvae*). Larvae used for plant infestations were placed as neonates on a rosette or lower stem leaf at a standardized position for each assay, and monitored mornings and evenings, during times outside of the main period of *Geocoris* spp. activity that occurs at midday. Plants in field experiments were either infested with *Manduca* spp. larvae as described above, or left uninfested (control). There were four infestations over both years of the experiment, denoted M1-M4 in *Figure 4*.

For measuring headspace GLVs in the field and for glasshouse assays, plants were treated with wounding and *M. sexta* OS (W+OS) as a standardized method to mimic *Manduca* spp. feeding. Pure OS collected from fourth to fifth instar *M. sexta* larvae from the Jena colony fed on WT plants was diluted 1:5 with distilled water before use; even 1000-fold diluted OS is still sufficient to cause most OS-elicited responses (*Schittko et al., 2000*). For field-grown plants, a similar, mature, non-senescent leaf was chosen from each plant; for glasshouse-grown plants, the two adjacent older leaves (nodes +1, +2) to the leaf undergoing a source-sink transition (node 0) on rosette-stage plants were used for *PI* and *LOX2* transcript quantification, and the +2 node of a separate set of bolting plants was used for

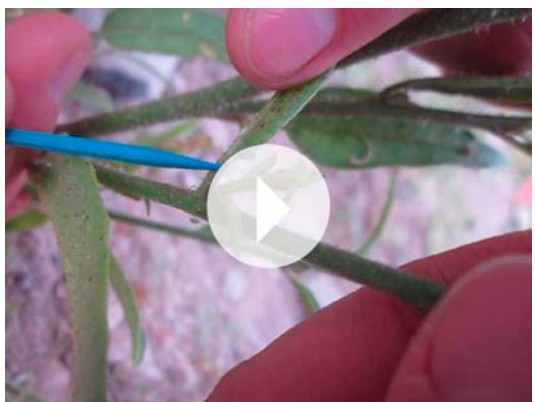

**Video 2**. On-plant assay, plant 2o, ir*PI*, June 18, 2011.

measuring headspace volatiles. The leaf chosen for treatment was wounded by using a fabric pattern wheel run over the adaxial surface to make six rows of holes in the lamina, three rows on either side of the midvein. 20 µL of 1:5 diluted OS were deposited on the adaxial surface and gently rubbed over the holes with a gloved finger. Control plants were left untreated.

## Plant tissue harvests and sample handling

For field-grown plants and glasshouse-grown *M. sexta*-fed plants, a similar, mature, non-senescent systemic (undamaged) leaf was chosen from each plant; for glasshouse-grown plants used to measure *PI* and *LOX2* transcripts, the leaves at nodes +1 and +2 (treated leaf positions) were harvested. Leaves were cut at the petiole and wrapped in a double layer of aluminum foil. In the field, harvested leaves were immediately frozen on dry ice insulated with ice packs frozen at −20°C; samples were stored at −20°C until transport to Jena on dry ice, where they were kept at −80°C until analysis. Leaves harvested from glasshouse-grown plants were flash-frozen in liquid nitrogen and kept at −80°C until analysis. All sample processing was carried out over liquid nitrogen until the addition of the extraction solvents. Prior to analysis, entire leaves were ground with a mortar and pestle and transferred to a 2 mL microcentrifuge tube for storage. For specific measurements, aliquots were weighed into microcentrifuge tubes containing two steel balls and finely ground in a GenoGrinder (SPEX Certi Prep) prior to extraction.

## Quantification of TPI activity

TPI activity was quantified in 100 mg of tissue from systemic leaves on *Manduca* spp.-infested plants using a radial diffusion assay as previously described (*van Dam et al., 2001*).

## Quantification of *PI* and *LOX2* transcripts

Leaf samples were from control plants or plants treated with W+OS. Treated leaf positions were harvested at the peak of transcript accumulation for *PI*, 12 hr (*Wu et al., 2006*) and *LOX2*, 14 hr (*Allmann et al., 2010*). Total RNA was extracted from leaves using the TRIzol reagent (Invitrogen), and a 0.5 µg aliquot of total RNA of each sample was reverse-transcribed using oligo(dT)$_{18}$ and RevertAid H Minus reverse transcriptase (Fermentas) following the manufacturer's instructions. Quantitative real-time PCR (qPCR) was performed with a Mx3005P Multiplex qPCR system (Stratagene) and the qPCR Core kit for SYBR Green I (Eurogentec). Transcripts were quantified using external standard curves for each gene. Elongation factor 1A (*EF1A*) transcript abundance in each sample was used to normalize total cDNA concentration variations. Samples of RNA used to make cDNA were pooled to the same dilution as in cDNA samples and run alongside cDNA in all qPCRs to control for gDNA contamination; no contamination was detected. The sequences of primers used for qPCR (*Kallenbach et al., 2010*; *Fragoso et al., 2011*) are provided in *Table 4*.

## Quantification of GLV pools in tissue

To assess qualitatively the GLV pools in leaf tissue from field-grown plants, and to determine appropriate amounts of leaf tissue and internal standard (IS) for GLV extraction, we extracted pooled samples from leaves collected June 6, 2011, from *M. sexta*-infested plants during infestation M3 (*Figure 4*). Each sample was pooled from all leaves collected from one genotype. Hexane (300 µL) was added to 100 mg tissue spiked with 3 µg tetralin as an internal standard (IS) and incubated by rotating at RT overnight. Samples were allowed to settle and 100 µL of water- and tissue-free hexane was transferred to a GC vial containing a 250 µL microinsert. Individual analytes were analyzed by a Varian CP-3800 GC-Saturn 4000 ion trap MS connected to a ZB5 column (30 m×0.25 mm i.d., 0.25 µm film thickness; Phenomenex). 1 µL of samples was injected by a CP-8400 autoinjector (Varian) onto the column with a 1:10 split ratio; the injector was returned to a 1:70 split ratio from 2 min after injection through the

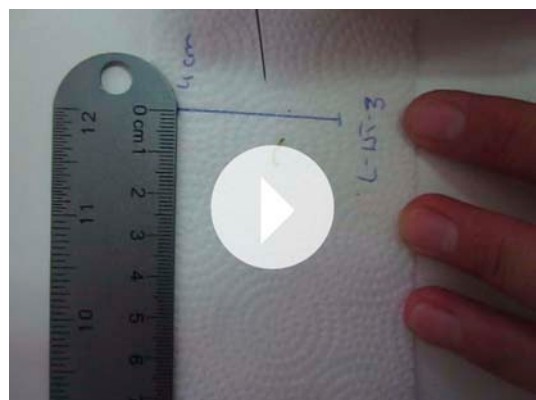

**Video 3**. Off-plant assay, replicate 3, WT, June 24, 2011.

end of each run. The GC was programmed as follows: injector held at 250°C, initial column temperature at 40°C held for 5 min, then ramped at 5°C/min to 185°C and finally at 30°C/min to 300°C, held for 0.17 min. Helium carrier gas was used and the column flow set to 1 mL/min. Compounds eluted from the GC column were transferred to the MS for analysis. The MS was programmed as follows: transfer line at 250°C, trap temperature 110°C, manifold temperature 50°C, source heater 200°C and scan range from 40 to 399 m/z at 1.33 spectra per second as previously described (*Schuman et al., 2009*). The identification of compounds was conducted by GC retention time compared to pure standards and mass spectra compared to standards and mass spectra databases, Wiley version 6 (Wiley) and NIST (National Institute of Standards and Technology) spectral libraries.

For the quantification of GLV pools in leaf tissue from field-grown plants, the hexane extraction protocol was adjusted based on GC-MS results from pooled samples (described above), and a GC-FID with a wax column was used for the quantitative analysis of extracts. Hexane (300 µL) were added to 50 mg tissue (N=10) spiked with 15 µg (*Z*)-hex-3-enyl acetate, a GLV not found in GC-MS samples, as an IS. The extraction proceeded as described above. Analytes were separated by Varian CP-3800 GC-FID connected to a ZB-Wax column (30 m×0.25 mm i.d., 0.25 µm film thickness; Phenomenex). 1 µL of samples was injected by a CP-8400 autoinjector (Varian) onto the column in a splitless mode; the injector was returned to a 1:70 split ratio 2 min after injection through the end of each run. The GC was programmed as follows: injector held at 230°C, initial column temperature at 40°C held for 7 min, then ramped at 5°C/min to 115°C and finally at 30°C/min to 250°C, held for 0.5 min. Helium carrier gas was used and the column flow set to 1 mL/min. Compounds eluted from the GC column were transferred to a Varian FID set at 250°C for analysis (airflow 300 mL/min, hydrogen 30 mL/min, nitrogen make-up gas 5 mL/min). Individual volatile compound peaks were quantified by peak areas using MS Work Station Method Builder and Batch Report software (Varian) and normalized to the peak area of the IS (*Z*)-hex-3-enyl acetate in each sample. Peak identification and quantification was done by comparison to standard curves of pure compounds in hexane. Compounds present in quantifiable amounts were (*Z*)-hex-3-en-1-ol, (*E*)-hex-2-enal and the IS (*Z*)-hex-3-enyl acetate.

## Relative quantification of GLVs and (*E*)-α-bergamotene in the plant headspace

For measurement of GLVs in the headspace of field-grown plants, intact leaves were harvested (N=3) and kept fresh by placing the petioles in microcentrifuge tubes filled with water. Immediately before each measurement, one leaf was treated with W+OS, and a 1 cm² disc was stamped out and placed in a 4 mL GC vial. After 15 min, the headspace in the vial was measured with a ZNose 4200 portable gas chromatograph with a 1 m DB5 column (Electronic Sensor Technology, Newbury Park, CA, USA) by inserting the ZNose inlet needle through the septum of the GC vial into the headspace. The program was as follows: valve set at 165°C, inlet at 200°C, trap at 250°C; 30 s sampling time, column ramped from 30°C to 190°C at 4°C/s, data collection for 20 s. Genotypes were analyzed in an alternating order within each replicate: first replicate 1 of all genotypes, then replicate 2, then replicate 3. Retention times of GLV aldehydes and alcohols, the most abundant GLV headspace components, were determined using pure standards.

For the analysis of GLVs in the headspace of glasshouse-grown plants, the +2 leaf was enclosed immediately after W+OS elicitation in a food-quality 50 mL plastic container (Huhtamaki) with an activated charcoal filter attached to one side for incoming air, and connected to self-packed Poropak Q filters containing 20 mg of Poropak (Sigma-Aldrich) packed with silanized glass wool and Teflon tubing in the column bodies (ARS, Inc.) as previously described (*Halitschke et al., 2000*; *Schuman et al., 2009*). Ambient air was pulled by vacuum pump for 3 hr through an activated charcoal filter, over the leaf in the trapping container, and through a Poropak Q filter connected by PVC tubing (Rotabilo) to a

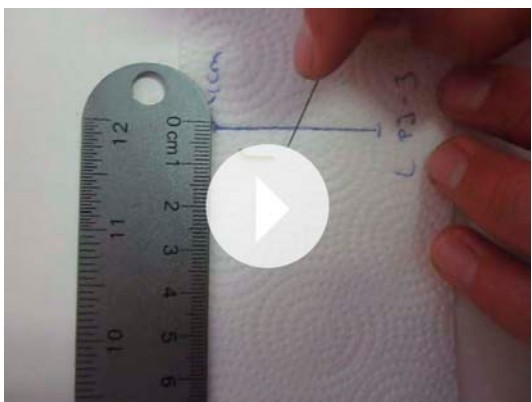

**Video 4**. Off-plant assay, replicate 3, ir*PI*, June 24, 2011.

custom-made valve manifold, as previously described (*Schuman et al., 2009*); the manifold was adjusted such that flow rates through traps were ca. 300 mL/min. After trapping, sampled leaves were excised at the base of the petiole, scanned, and the leaf area was measured in comparison to a 1 cm² standard (SigmaScan 5.0; Systat Software Inc.) for normalization of volatile emission to cm² leaf area. Poropak Q filters were wrapped in aluminum foil and stored at −20°C until elution of volatiles with 250 µL dichloromethane (Sigma-Aldrich).

Immediately prior to elution, each filter was spiked with 320 ng of tetralin internal standard (IS) in hexane (Sigma-Aldrich). Filters were eluted into a GC vial containing a 250 µL glass insert. Samples were analyzed by a CP-3800 GC Varian Saturn 2000 ion trap MS (Varian) connected to a polar ZB-wax column (30 m×0.25 mm i.d., 0.25 µm film thickness; Phenomenex). 1 µL of samples was injected by a CP-8200 autoinjector (Varian) onto the column in a splitless mode; the injector was returned to a 1:70 split ratio from 2 min after injection through the end of each run. The GC was programmed as follows: injector held at 230°C, initial column temperature at 40°C held for 3 min, then ramped at 5°C/min to 180°C and finally at 10°C/min to 240°C, held for 1 min. Helium carrier gas was used and the column flow set to 1 mL/min. Eluted compounds from the GC column were transferred to the MS for analysis. The MS was programmed as follows: transfer line at 230°C, trap temperature 150°C, manifold temperature 80°C and scan range from 40 to 399 m/z at 1.33 spectra per second as previously described (*Schuman et al., 2009*). Individual volatile compound peaks were quantified by peak areas of two specific and abundant ion traces per compound using MS Work Station Data Analysis software (Varian) and normalized by the 104+132 ion trace peak area of the IS (tetralin) in each sample. The identification of compounds was conducted by GC retention time compared to pure standards and mass spectra compared to standards and mass spectra databases, Wiley version 6 (Wiley) and NIST (National Institute of Standards and Technology) spectral libraries. In 3 hr headspace samples we detected (*Z*)-hex-3-en-1-ol, (*Z*)-hex-3-en-1-ol, (*E*)-hex-2-en-1-ol (forms from (*E*)-hex-2-enal on filters over trapping periods longer than 20 min), (*Z*)-hex-3-enyl acetate, (*Z*)-hex-3-enyl butanoate, (*Z*)-hex-3-enyl isobutyrate, and (*Z*)-hex-3-enyl propanoate.

The collection of (*E*)-α-bergamotene from the headspace of glasshouse-grown plants and its extraction from Poropak Q filters was carried out as for GLVs, except that (*E*)-α-bergamotene was collected 24–32 hr after W+OS treatment of the leaf. Eluted samples were analyzed by an HP 6890 GC-5973 quadropole MS (Hewlett-Packard) connected to a nonpolar DB-5ms column (30 m×0.25 mm i.d., 0.25 µm film thickness; Agilent). 1 µL of samples was injected by a HP 7683 autoinjector (Hewlett-Packard) onto column in a splitless mode; the injector was purged at 50 mL/min 1.5 min after injection and switched to gas saver mode (20 mL/min) from 10 min through the end of each run. The GC was programmed as follows: injector held at 230°C, initial column temperature at 40°C held for 2 min, then ramped at 5°C/min to 165°C and finally at 60°C/min to 300°C, held for 2 min. Helium carrier gas was used and the column flow set to 2 mL/min. Eluted compounds from GC column were transferred to the MS for analysis. The MS was programmed as follows: source at 230°C, quad temperature 150°C, and scan range from 33 to 350 m/z at 4.49 spectra per second. (*E*)-α-Bergamotene was quantified by peak area using the ion trace 119 m/z in Chemstation software (Agilent) and normalized by the 104 ion trace peak area of the IS (tetralin) in each sample. The identification of (*E*)-α-Bergamotene and tetralin IS was conducted by GC retention time and mass spectra compared to mass spectra of known standards as previously described (*Schuman et al., 2009*).

### *Manduca* spp. bioassays

One or two larvae were placed on plants at a time for each assay, depending on the number available, and were equally distributed among plants as described under '*Manduca* spp. infestation and W+OS treatment of plants'. However, for infestation M3 (*Figure 4*), we staggered the infestation of different plant genotypes to accommodate differences in plant growth: WT and ir*PI* plants were initially larger and therefore went into the field on average earlier than ir*LOX2* and hemi-ir*LOX2* plants, so that all

**Table 4.** Primers used for quantitative PCR (SYBR Green)

| Gene | Forward primer sequence (5′-3′) | Reverse primer sequence (5′-3′) | Citation |
|------|----------------------------------|----------------------------------|----------|
| *PI* | TCAGGAGATAGTAAATATGG | ATCTGCATGTTCCACATTGC | *Fragoso et al. (2011)* |
| *LOX2* | TTGCACTTGGTGTTTGAGATGGT | TTAGTAGAAAATGAGCACCACAA | *Kallenbach et al. (2010)* |

plants were planted at a similar size, which is important for even establishment. We therefore re-infested WT and ir*PI* plants earlier after M1, to allow ir*LOX2* and hemi-ir*LOX2* plants to catch up in their growth to WT and ir*PI* before re-infestation. However, we then left *M. sexta* larvae on ir*LOX2* and hemi-ir*LOX2* as long as on WT and ir*PI*, and we used a combination of *Geocoris* spp. counts and additional predation assays to make sure that differences in *Geocoris* spp. predation were not due to this staggering of infestation (see 'Results: *Geocoris* spp. preferentially predate from GLV-perfumed or -emitting plants').

*Manduca* spp. behavior, predation, and growth assays were conducted with first- and second-instar larvae, except infestations M1 2010 and M4 in 2011, in which larvae were reared from the first through fifth instars on plants; and egg predation assays, in which *M. sexta* eggs were used.

Larvae used in the off-plant mock predation assays were hatched on the appropriate *N. attenuata* genotype (WT or ir*PI*) and hatching was monitored three times per day (morning, noon, evening) so that the mock predation assay could be timed to 48 hr after larvae hatched. A protocol of the mock predation assays is given in *Figure 10*, and *Videos 1–4* depict on-plant (1 and 2) and off-plant (3 and 4) behavioral assays. Larvae for off-plant mock predation assays were kept in aerated plastic boxes on cut leaves over moist paper towels. Leaves were refreshed twice daily and were kept fresh by placing the petioles in water in 1.5 mL microcentrifuge tubes which were closed around the petiole with Parafilm (Pechiney Plastic Packaging Company). Larval growth was measured as increases in body length (in millimeters) using calipers or a small, flexible, transparent plastic ruler.

## Predation assays

Predation rates were recorded for larvae placed on plants as described above, or for two eggs per plant fixed with droplets of α-cellulose glue (*Kessler and Baldwin, 2001*) to the underside of a rosette or lower stem leaf at a standardized position. For egg predation assays, a wild *Manduca* spp. larva was enclosed on a nearby leaf to ensure continual GLV emission: a clip cage was closed around the larva to make it inaccessible to *Geocoris* spp. predators. Predation was monitored mornings and evenings. Larvae were considered to be predated when either the larva was missing over multiple days, but clear *Manduca* spp. feeding damage was present, or when the predated larval carcass was found (*Figure 5*). Mortality was defined as the total number of missing larvae. Eggs were considered predated when the eggshell was empty but intact except for a small hole which characterizes the typical damage caused by *Geocoris* spp. feeding; eggs occasionally collapse during *Geocoris* spp. predation, but collapsed eggs were not counted unless the eggs were mostly or fully empty and with a visible hole (*Figure 5*).

## GLV supplementation

During infestation M2, GLVs were added back to ir*LOX2* and hemi-ir*LOX2* headspaces by placing a cotton swab adjacent to the *M. sexta*-infested leaf and adding ca. 20 µL of lanolin paste, measured with a seed spoon, containing a mix of pure GLVs representative of the *M. sexta*-fed headspace and dissolved in hexane (*Table 1*) (*Allmann and Baldwin, 2010*) to the cotton swab. Cotton swabs bearing ca. 20 µL of lanolin paste with hexane as a control were placed next to *M. sexta*-infested leaves of WT and ir*PI* plants. Lanolin pastes were regularly refreshed by adding 20 µL in the early afternoon and in the morning. Placing GLVs next to, rather than on the leaf ensured that the supplemented headspace would not be altered by plant metabolism, and that we could terminate the supplementation by removing the cotton swabs.

## *Geocoris* spp. counts

Field plots were monitored daily for *Geocoris* spp. presence during the experiments in 2010 and 2011; both *G. pallens* and *G. punctipes* were present in 2011, but most individuals observed on and around plants were *G. pallens*. Soon after the first *Geocoris* spp. sightings in May 2011 (before

the first infestation, M2), *Geocoris* spp. populations in the immediate vicinity of experimental plants were monitored every 2–3 days by counting individuals. Counts were conducted during the main period of *Geocoris* spp. activity in the early afternoon, by at least two observers in parallel, in order to complete the count around all *Manduca* spp.-infested and control plants within 20–30 min. Each observer proceeded by looking at a focal plant and its immediate vicinity for 15 s and then quickly inspecting the rosette leaves; all *Geocoris* spp. adults and nymphs seen on, under, or within 5 cm around the rosette of the plant during this time were counted. Observers moved in synchrony with each other from one end of the field plot to the other, in this way counting predators around plants which had not yet been disturbed.

## Plant growth and reproduction

Plant size (rosette diameter, stem length and branching) was monitored at the end of infestation M1 in 2010 and from the beginning of infestation M2 in 2011 (*Figure 4*): rosette diameter was measured as the maximum diameter found by gently laying a ruler over the rosette; stem length was measured from the base of the stem to the tip of the apical inflorescence by placing a ruler beside the stem; and all side branches 5 cm or longer were counted. Reproductive output was monitored by counting the number of closed flowers removed every 2–3 days (before they opened) from the beginning of flowering, and by counting numbers of closed flowers and buds 2 mm or larger at the end of infestation M1 in 2010, and during all infestations in 2011. All growth and reproduction data were analyzed for differences in control versus *M. sexta*-infested plants within each genotype (statistics *Table 3*). Because size-matched plants had been planted over one week in 2011, growth and reproduction data from plants in 2011 were organized by the number of days since planting for comparison between genotypes (*Figure 6*).

## Removal of flowers during infestations M1 and M3, and of reproductive meristems prior to infestation M4

Flowers were removed and counted periodically over the first 10 days of infestation M1 and during infestation M3 (*Figures 4 and 6*, statistics *Table 3*), in order to track plant reproduction while avoiding ripe seed capsules: the distribution of ripe seed is not permitted for genetically modified plants. For infestation M4 in 2011 (*Figure 4*), instead of regularly removing flowers, we removed all reproductive meristems by cutting inflorescences, which contained mostly buds, at their base, so that we would be able to follow a new set of reproductive meristems through to seed set without incurring ripe seed. Because plants were matched prior to M4 and had the same number of buds and of side branches (which usually terminate in inflorescences) (*Figure 8*, see 'Discussion'), all plants were similarly affected by the removal of reproductive meristems.

## Herbivore damage and plant health

Photographs were taken of entire plants and *M. sexta*-damaged leaves during infestations M2 and M3 in 2011. Damage caused by *M. sexta* larvae was rated from photographs by an independent observer with no knowledge of plant identity. Total percent canopy damage due to *M. sexta* was rated as 1, 2, 3, or 4 using the damage index in *Figure 7*.

We monitored herbivore attack to determine whether GLV-silenced plants suffered different amounts of herbivore damage, which could influence fitness measurements. The naturally occurring herbivore community on plants in 2010 and 2011 comprised mirids (*Tupiocoris notatus*) and noctuid larvae; in 2011 grasshoppers (*Trimerotropis* spp.) and flea beetles (*Epitrix* spp.) were also present. Total canopy damage due to herbivores occurring naturally on the field plot was quantified prior to the infestations in 2010 and 2011 and again during infestation M3 in 2011. Damage was calculated by identifying damage from specific herbivores according to their characteristic feeding patterns, counting the number of leaves per plant (small leaves were counted as 1/5 to 1/2 of a leaf based on leaf area and large leaves were counted as 1 leaf), estimating the total percentage of leaf area damage due to each herbivore, and dividing the total leaf area damage from each herbivore by the total number of leaves. Leaf area damage was estimated in categories of 1%, 5%, 10%, 15%, and so on, in steps of 5%. All such damage estimates were made by MCS or KB, who first practiced quantifying damage together until they consistently arrived at the same numbers.

As part of matching plants prior to infestation M4 in 2011, plant health was rated on a scale of 1 (dead) to 5 (healthy) using the index in *Figure 7*.

## Statistical analyses

Fisher's exact tests were conducted using a macro (J H Macdonald, http://udel.edu/~mcdonald/statfishers.html) for Excel (Microsoft). All other statistical analyses were conducted with SPSS 17.0 (IBM). Count data were analyzed either by Fisher's exact tests (independent values) or by Friedman tests (repeated measures). Levene's test for homogeneity of variance was performed prior to all t-tests and ANOVAs and when necessary, data were $\log_2$ transformed (volatile and transcript data), square root transformed (count data) or arcsin transformed (herbivore damage data) to meet requirements for homogeneity of variance. Parametric data were compared using ANOVAs, MANOVAs, or repeated-measure ANOVAs followed by Scheffe *post hoc* tests. If variance was not homogeneous following transformation, data were compared using Kruskal-Wallis tests (for multiple comparisons) or Mann-Whitney U-tests (for two-way comparisons) and Bonferroni p-value corrections were used to correct for nonparametric multiple comparisons. For Kruskal-Wallis tests and Mann-Whitney U-tests, a Monte Carlo algorithm was used with 10,000 permutations and a 95% confidence level.

## Acknowledgements

We thank S Allmann for conducting egg predation assays in 2010; C Diezel, M Erb, M Kallenbach, D Kessler, D Marciniak, M Stanton, and A Steppke for help with field work; S Weigl for rating *M. sexta* damage levels; J Gershenzon for the use of the HP GC-MS; M Erb, J Gershenzon, D Heckel, M Kallenbach, A Kessler, A Steppke, and A Weinhold for comments on the manuscript; Brigham Young University for the use of their Lytle Preserve field station; and APHIS for constructive regulatory oversight.

## Additional information

### Competing interests

ITB: Senior Editor, *eLife*. The other authors have declared that no competing interests exist.

### Funding

| Funder | Author |
| --- | --- |
| Max Planck Society | Meredith C Schuman, Kathleen Barthel, Ian T Baldwin |

The funder had no role in study design, data collection and interpretation, or the decision to submit the work for publication.

### Author contributions

MCS, Conception and design, acquisition of data, analysis and interpretation of data, and drafting or revising the article; KB, Conception and design, acquisition of data, analysis and interpretation of data, and drafting or revising the article; ITB, Conception and design, acquisition of data, analysis and interpretation of data, and drafting or revising the article.

## Additional files

### Major datasets

The following datasets were generated

| Author(s) | Year | Dataset title | Dataset ID and/or URL | Database, license, and accessibility information |
| --- | --- | --- | --- | --- |
| Schuman MC, Barthel K, Baldwin IT | 2012 | Herbivory-induced volatiles function as defenses increasing fitness of the native plant Nicotiana attenuata in nature | http://dx.doi.org/10.5061/dryad.gs45f | Available at Dryad Digital Repository under a CC0 Public Domain Dedication |

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
