## [Decision Letter]

Your manuscript “Herbivory-induced volatiles function as defenses increasing plant fitness in nature” has been reviewed. The following individual(s) responsible for the peer review of your submission want(s) their identity to be known: Detlef Weigel (reviewing editor); Joerg Bohlmann (reviewer).

We are happy to tell you that a revised manuscript shall be acceptable for *eLife*. Here are the comments:

The hypothesis outlined in the title of this paper, that indirect defences in the form of herbivory-induced volatiles increase plant fitness, is being continuously discussed in the literature, without there being much definitive evidence for this phenomenon occurring in nature. It is a difficult hypothesis to test, but the authors of this work make a big step toward providing an answer.

In field studies over two seasons (with both wild and lab-reared predators) complemented with lab experiments, the authors examined how emission of volatile organic compounds (VOCs) and a protease inhibitor (TPI) that reduces digestibility of plant tissue affect herbivore growth and plant fitness, as measured by flower buds and seed set in tobacco. The experimental approach was to reduce VOCs by silencing the *LOX2* gene, and TPI activity by silencing the PI gene.

The authors clearly show that plants that emit less VOCs attract fewer *Geocoris* predators to *Manduca* herbivores (Fig. 2), and that *Manduca* mortality is reduced in the presence of *Geocoris* (Fig. 3A). The result is that fewer flowers and flower buds are produced, but only in the presence of *Geocoris* (Fig. 3B).

The authors are to be commended for reporting a comprehensive set of measurements, including ones that are not easily explained or fit the hypotheses examined. This makes unfortunately in several places for difficult reading. In addition, the paper is extremely densely written and difficult to follow for those unfamiliar with the research program of the authors. To make the content of the work more accessible, the paper should be presented with separate sections for introduction, results, and discussion, and subsections for the results. In addition, the supplementary figures as well as the supplementary narrative should be integrated into the main text, and the data should be better and more consistently presented in the figures. For example, the colour code for different genotypes in Fig. S1-3 and 6A,B should also be applied in all other figures.

Some essential information is only mentioned in passing, e.g., that homozygous ir*LOX2* lines “… did not suffer reduced growth or reproduction from *Manduca sexta* feeding … and that they were too small …”. This must be explained, and better justification must be provided for focussing on the hemizygous lines, especially because it is not entirely clear how the effects on VOCs compare between hemi- and homozygous plants.

Other important points: (i) The statement in the Introduction (beginning “Although GLVs are released upon mechanical damage…”) must be at least changed to: “… the oral secretions of *Manduca* convert 3-(Z)-GLVs to the 2-(E)-structures, …” (This is because this is a rearrangement, not an isomerisation). (ii) The authors should consider whether the experiments in Fig. 4 regarding the effects of TPI on *M. sexta* behaviour, though interesting, distract from the rest of the work. Perhaps they could be omitted from this manuscript?

A final remark: it is a pity that, because of global attitudes to genetically modified plants, the authors could not evaluate ripe seeds.

---

## [Author Response]

Reviewer comments are in italics.

*The authors are to be commended for reporting a comprehensive set of measurements, including ones that are not easily explained or fit the hypotheses examined. This makes unfortunately in several places for difficult reading. In addition, the paper is extremely densely written and difficult to follow for those unfamiliar with the research program of the authors. To make the content of the work more accessible, the paper should be presented with separate sections for introduction, results, and discussion, and subsections for the results. In addition, the supplementary figures as well as the supplementary narrative should be integrated into the main text, and the data should be better and more consistently presented in the figures. For example, the colour code for different genotypes in Fig. S1-3 and 6A,B should also be applied in all other figures*.

We have now organized the manuscript into an introduction, Materials and Methods, results, and discussion, with subsections in the Methods, Results, and Discussion sections following the order of the figures and tables. In particular, we have expanded the explanation of the data behind new Figures 8 and 9 (infestations M1 and M4).

We have also implemented consistent color codes for the genotypes, and consistent pattern codes for control vs *Manduca* spp.-infested or W+OS-treated plants, and a more consistent format for the figures. We have also merged some figures which were formerly separate but described the same results or sets of assays.

A note: in the interest of clarity, we now include reports of statistical analyses in the text and not only in figure captions or tables. Perhaps it could be considered to convert these in-text statistical reports to mouse-over hypertext in the online version of the paper, since the long parantheticals interrupt the flow of the text.

*Some essential information is only mentioned in passing, e.g., that homozygous irLOX2 lines “… did not suffer reduced growth or reproduction from Manduca sexta feeding … and that they were too small …”. This must be explained, and better justification must be provided for focussing on the hemizygous lines, especially because it is not entirely clear how the effects on VOCs compare between hemi- and homozygous plants (see minor comments below)*.

We have tried to clarify our description of the GLV pools and emissions from the ir*LOX2* and hemi-ir*LOX2* plants. It is clearly stated in the Materials and Methods and in the Results that the hemi-ir*LOX2* plants contain an inactive ir*PI* construct, and the possible explanations for this construct's inactivity are discussed in a new section of the discussion, “WT *levels of TPIs in hemi-ir*LOX2* plants are likely due to gene dosage effects”*.

*Other important points: (i) The statement in the Introduction (beginning “Although GLVs are released upon mechanical damage…”) must be at least changed to: “… the oral secretions of *Manduca* convert 3-(Z)-GLVs to the 2-(E)-structures, …” (This is because this is a rearrangement, not an isomerisation)*.

We have changed this sentence to “Although GLVs are released upon mechanical damage, the oral secretions (OS) of *M. sexta* convert 3-(Z)-GLVs to the 2-(E)-structures, resulting in greater *Geocoris* spp. predation than the damage-induced (Z):(E) ratio”.

*(ii) The authors should consider whether the experiments in Fig. 4 regarding the effects of TPI on M. sexta behaviour, though interesting, distract from the rest of the work. Perhaps they could be omitted from this manuscript*?

We recognize that this is a long, complex, and data-heavy manuscript. However, we feel that the behavior data is essential to the story, because it is the only effect we found of TPIs, and it is a novel effect which is highly relevant for the efficiency of indirect defense. Both because of the experimental designs presented in the paper, and because of the implications of these results, we feel they belong as part of this story. Also, if these data were not included, we would have to re-analyze all experiments without the ir*PI* line, which we feel is artificial. The removal of these data would only save one figure and 3.5 paragraphs of text.

*A final remark: it is a pity that, because of global attitudes to genetically modified plants, the authors could not evaluate ripe seeds*.

We appreciate the reviewers' sympathy on this point!